# PhaseCoder: Microphone Geometry-Agnostic Spatial Audio Understanding for Multimodal LLMs

Artem Dementyev [* 1]  Wazeer Zulfikar [* 2 1]  Sinan Hersek [3]  Pascal Getreuer [1]  Anurag Kumar [1]  Vivek Kumar [1]

## Abstract

Current multimodal large language models (LLMs) process audio as a mono stream, ignoring the rich spatial information essential for embodied AI. Conversely, existing spatial audio models are constrained to fixed microphone geometries, preventing their deployment across diverse devices. We present **PhaseCoder**, a transformer-only spatial audio encoder that is inherently agnostic to microphone geometry. By taking raw multichannel audio and microphone coordinates as inputs, PhaseCoder performs accurate localization and produces robust spatial embeddings. We demonstrate that the Gemma 3n LLM can be fine-tuned to process and reason over the "Spatial Audio Tokens" produced by our encoder. PhaseCoder achieves state-of-the-art results on microphone-invariant localization benchmarks and, for the first time, enables an LLM to perform complex spatial reasoning and targeted transcription tasks from an arbitrary microphone array. Our models are publicly released at https://github.com/google-deepmind/phasecoder.

## 1. Introduction

While current multimodal large language models (LLMs) understand spoken words in a single-channel audio stream, they operate largely "audio-blind" to the rich information that spatial sound provides about the real world. In contrast, spatial audio understanding is fundamental to many living organisms; it enables humans to communicate in noisy environments (the "cocktail party effect") and owls to pinpoint the movement of prey, such as mice, deep in the snow (Takahashi, 2010). Similarly, spatial awareness is critical for embodied intelligence in robotics and next-generation AI assistants. Thanks to the proliferation of inexpensive, small MEMS omnidirectional microphones, most modern devices—from home assistants to wearables—are already equipped with microphone arrays capable of capturing this spatial data. However, despite the ubiquity of this hardware, its potential for spatial audio understanding remains **largely untapped**.

Recent research has shown that LLMs can be fine-tuned for understanding and reasoning with spatial audio (Zheng et al., 2024; Tang et al., 2024). However, these approaches lack a universal spatial representation. Because of the vast diversity of microphone geometries, a separate encoder must be trained for every new device, and the fine-tuned LLM is then locked to that exact geometry. This greatly limits the main generalization advantage of LLMs. Using microphone-agnostic representations such as Ambisonics is an option (Tang et al., 2024); however, recording them requires specialized hardware, such as a tetrahedral microphone array[1] or a high-order spherical array[2], which are often too bulky for consumer devices. Alternatively, arbitrary microphone arrays can be converted to Ambisonics (Gayer et al., 2024) or custom array-agnostic formats, but this requires device-specific beamformers. Furthermore, an additional encoder is still required to convert Ambisonics into spatial audio embeddings. Separately, recent research has shown the possibility of training microphone-geometry-agnostic models that operate on raw audio for sound localization (Baek et al., 2025). However, these approaches have not yet been integrated with LLMs for complex tasks such as question answering and reasoning.

In this paper, we bridge the gap between universal spatial audio encoding and its understanding by LLMs. To do so, we develop *PhaseCoder*, a microphone-geometry-agnostic spatial audio encoder for LLMs. The *PhaseCoder* name stems from the fact that spatial information is largely encoded in small phase differences between microphones in an array, as shown in Figure 2. It is designed to work with omnidirectional microphones (e.g., MEMS), as they are ubiquitous in consumer electronics. To ground the localization data to physical geometry, the coordinates of each

---

[*]Equal contribution  [1]Google DeepMind, Cambridge, USA [2]Media Lab, MIT, Cambridge, USA [3]Google AR, Seattle, WA. Correspondence to: Artem Dementyev <artemd@google.com>.

*Proceedings of the 43rd International Conference on Machine Learning*, Seoul, South Korea. PMLR 306, 2026. Copyright 2026 by the author(s).

---

[1]e.g., the Sennheiser AMBEO VR Mic or Røde NT-SF1.
[2]e.g., the mh Eigenmike em32.

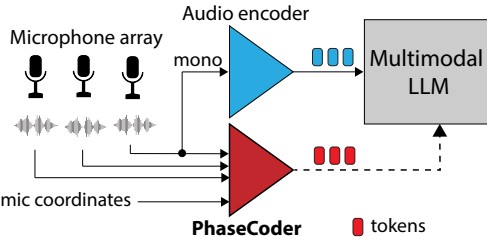

*Figure 1.* Overall system diagram. The spatial audio tokens are injected into the LLM alongside the existing "mono" audio tokens.

microphone are consumed by PhaseCoder. In many devices, these coordinates can be obtained from the API, such as in the Android OS[3].

The contributions of this paper are as follows:

- We develop *PhaseCoder*, a novel, transformer-only spatial audio encoder that is agnostic to microphone geometry. By taking microphone coordinates and multichannel audio as input, the encoder performs denoising, spatial localization, and distance estimation. It achieves state-of-the-art results among microphone-invariant encoders on multiple localization benchmarks and is computationally efficient.
- We demonstrate that the encoder produces robust, microphone-independent representations—"spatial audio tokens"—which are suitable for downstream tasks. We validate this by fine-tuning the Gemma 3n LLM to understand and reason with these spatial tokens **alongside** its existing mono audio tokens (Figure 1).
- We investigate and evaluate novel capabilities of the fine-tuned model, specifically spatial reasoning and the ability to transcribe speech from a target direction.

**Conflict of Interest Disclosure.**   The authors are employed by Google, which leads the development of the Gemma and Gemini models evaluated in this paper.

## 2. Previous Work

### 2.1. Microphone-Agnostic Localization

Classical audio localization algorithms typically operate by using cross-correlation between pairs of microphones (Rui & Florencio, 2003) or a steered beam response (Valin et al., 2004). Such algorithms are "microphone-geometry-agnostic" in the sense that different geometries can be given as algorithmic input parameters. However, classical algorithms struggle to adapt in noisy real-world conditions, whereas machine learning approaches have been shown to be more robust. Convolutional neural network (CNN)

---

[3]https://developer.android.com/reference/android/media/MicrophoneInfo

architectures have been shown to be especially effective in processing multichannel audio (Chakrabarty & Habets, 2017; Yalta et al., 2017). Another active area is Sound Event Localization and Detection (SELD), which aims to combine sound localization and classification into a single model (Adavanne et al., 2018). These models are typically trained on a single microphone geometry and do not work if the geometry or the number of microphones changes.

Recently, new learning-based approaches have emerged that are microphone-array-invariant. Neural-SRP (Grinstein et al., 2023) is the first instance of such a model. The model extracts pairwise features from the cross-correlation between microphone pairs, along with the microphone coordinates. The features are summed and processed by a global multilayer perceptron (MLP) network to determine the direction of arrival. Another approach (Schwartz et al., 2023) is to implicitly integrate the microphone geometry into spatial images based on pairwise features. A vision-inspired U-Net model then estimates the direction of arrival. Closest to our work is GI-DOAEnet (Baek et al., 2025), which improves localization by using latent features and positional embeddings derived from the microphone coordinates.

Previous microphone-invariant models have focused on audio localization tasks and have not been used to produce embeddings for LLMs. While Vision Transformers (ViT) (Dosovitskiy et al., 2020) and Audio Spectrogram Transformers (AST) (Gong et al., 2021) have been effective for creating embeddings, their transformer encoder architecture has not yet been applied to the task of microphone-invariant spatial localization, which has thus far been dominated by more complex hybrid architectures.

### 2.2. Representations of Spatial Audio for LLMs

Speech-focused mono audio understanding is present in many state-of-the-art multimodal LLMs. For example, Gemma 3n (Team et al., 2025) is pretrained using the mono USM (Universal Speech Model) (Zhang et al., 2023) encoder, mostly for transcription tasks. Furthermore, audio-specific foundational models that understand speech and other sounds have been demonstrated (Goel et al., 2025).

Recognizing an absence of spatial audio research, recent papers have begun to investigate how spatial audio understanding can be added to LLMs. In most cases, the successful approach is to fine-tune a large language model with some kind of spatial audio representation and a text output target. In an early example (Tang et al., 2024), Whisper encoder embeddings were combined with intensity vectors and used to fine-tune a Llama model on spatial speech tasks. Closest to our architecture, BAT (Zheng et al., 2024) introduced Spatial-AST, a dedicated encoder to produce soft spatial embeddings. A Llama V2 model was fine-tuned with those embeddings to understand spatial

audio. However, BAT did not extend to arbitrary spatial audio; it worked on binaural (2-channel) audio only and did not process speech. Another smart-glasses-centric approach in Directional-SpeechLlama (Xie et al., 2025) and M-BEST-RQ (Yang et al., 2025) is to use Non-Linearly Constrained Minimum Variance (NLCMV) beamforming to capture spatial audio information. While beamformer output is microphone-geometry-agnostic for ego-centric devices (e.g., glasses), the beamformers do not generalize to arbitrary devices. In another approach, ELSA (Devnani et al., 2024) was pre-trained from scratch using contrastive learning between spatial audio and text embeddings. Overall, previous works have not demonstrated microphone-geometry-agnostic encoders for use with LLMs.

### 2.3. Editing and Generation

Recent spatial audio generation and editing systems further motivate the need for rich spatial representations. SpatialSonic (Sun et al., 2025) conditions a latent diffusion model on spatial-aware encoders and azimuth state matrices. Similarly, ViSAGe (Kim et al., 2025) autoregressively generates first-order Ambisonics from silent video using directional and visual guidance. Furthermore, systems like SALM (Hu et al., 2025) and SmartDJ (Lan et al., 2026) integrate language models into the editing process; SALM learns structured audio embeddings for flexible editing, while SmartDJ decomposes high-level instructions into atomic spatial edit operations executed by a diffusion model. These works suggest that universal spatial audio encoders could serve not only for reasoning but also as control signals for generation and editing pipelines.

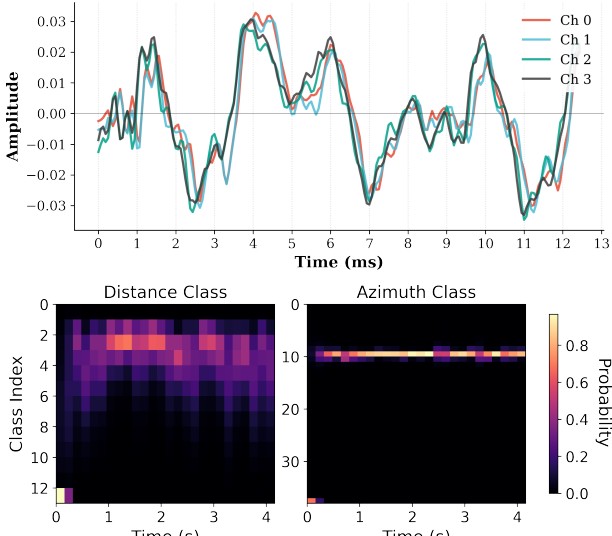

*Figure 2.* Visualization of input (raw audio) and classification heads of PhaseCoder. Top: Slice of four microphone channels raw audio, showing phase differences, which can be used to determine the direction of arrival. Bottom: Class probability for distance and azimuth over time.

## 3. Methods

### 3.1. PhaseCoder: Spatial Audio Encoder

This section describes the architecture and training of PhaseCoder. The overall architecture is shown in Figure 3.

**Input Audio Features** We first process the raw audio from each of the $C$ microphone channels. We compute the Short-Time Fourier Transform (STFT) using a 256-sample Hann window with a 50% overlap (128-sample hop). The audio sampling rate is 16 kHz. This process yields 33 STFT frames (a total of 250 ms) for the model input, ultimately producing a spatial audio token.

For a real-valued input, an STFT with a window size of $N = 256$ produces $N/2 + 1 = 129$ complex frequency bins. We extract the magnitude and phase for each of these 129 bins and concatenate them, resulting in an initial feature vector of dimension $129 + 129 = 258$ [4]. This 258-dimensional vector is then passed through a linear projection layer to produce the final $D = 256$-dimensional embedding.

We define a **patch** as this final $D$-dimensional embedding for a single microphone at a single time frame. The frame ($F = 33$) and channel ($C$) dimensions are then flattened to create a single input sequence of length $L = F \times C$. The final input tensor passed to the encoder has the shape $B \times L \times D$, where $B$ is the batch size.

**Positional Embeddings** Since the flattened patch embeddings ($B \times L \times D$) contain no explicit temporal or spatial information, we inject this by summing three distinct types of positional embeddings. We use fixed embeddings rather than learned ones to support a variable number of microphones ($C$).

1. **Sequential Embedding:** A standard 1D sinusoidal positional embedding is added to each patch based on its position in the flattened sequence (from 1 to $L$).
2. **Frame Embedding:** To help the model group information from the same time step, a separate sinusoidal embedding based on the frame index ($f \in [1, F]$) is added. This embedding is identical for all patches belonging to the same frame.
3. **Microphone Positional Embedding:** To encode the array geometry, we add a microphone-specific position embedding to each patch. We adapt the phase modulation embeddings introduced in GI-DOAEnet (Baek et al., 2025), which we found were more stable to train than the proposed frequency modulation embeddings.

To calculate microphone positional embeddings, first, Cartesian microphone coordinates are converted to spherical coordinates relative to the centroid, as described by the equations

---

[4]Practically, 2 of the 258 dimensions are unnecessary, as they are DC and Nyquist bins, which are real-valued.

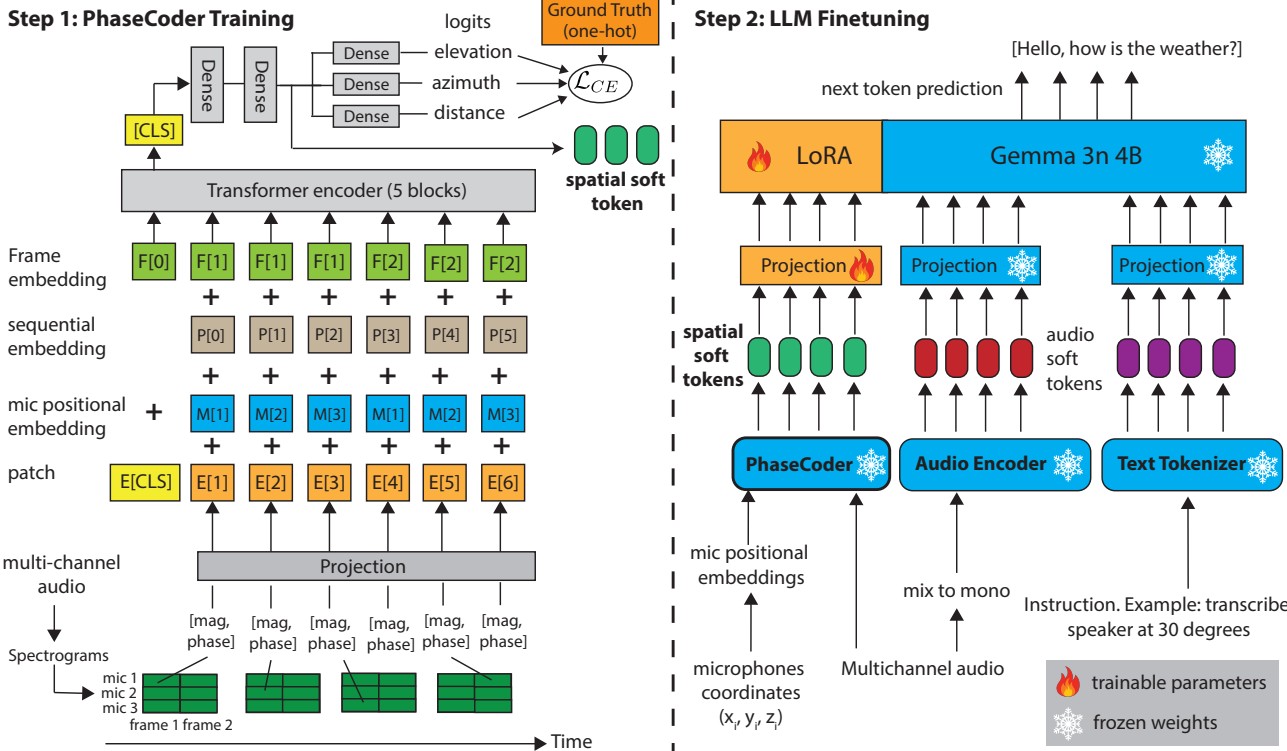

*Figure 3.* Detailed model architecture. Left: PhaseCoder model architecture and training. The spatial encoder is trained to predict discretized spatial coordinates (azimuth, elevation, distance) using a cross-entropy objective ($\mathcal{L}_{CE}$). For clarity, only two frames from three microphones are shown. Right: Adding spatial tokens to the Gemma 3n language model. The PhaseCoder input is added to the existing input pillars (mono audio and text). Down-mixing to mono can be done with various techniques, such as averaging channels or selecting one channel.

in Appendix A.

Second, the positional embedding $\mathcal{P}_i$ for each microphone channel (index $i$) is calculated. This positional embedding is a $D = 256$-dimensional vector, created as:

$$\mathcal{P}_i = \alpha r_i \begin{bmatrix} \cos(2\pi\beta\mathbf{v} + \theta_i) \\ \sin(2\pi\beta\mathbf{v} + \theta_i) \\ \cos(2\pi\beta\mathbf{v} + \phi_i) \\ \sin(2\pi\beta\mathbf{v} + \phi_i) \end{bmatrix},$$

where $\alpha = 7.0$ and $\beta = 4.0$ are constants from the original GI-DOAEnet, $r_i$ is the radius, $\theta_i$ is the elevation, and $\phi_i$ is the azimuth.

The base vector $\mathbf{v}$ is defined as a uniform sequence:

$$\mathbf{v} = \frac{4}{D}\left[0, 1, \ldots, \frac{D}{4} - 1\right]^\top,$$

where $D = 256$ is the embedding dimension.

**Backbone** We employ a Transformer encoder backbone, similar in principle to the ViT architecture (Dosovitskiy et al., 2020). The model's embedding dimension is $D = 256$. The backbone consists of 5 standard Transformer

blocks, each containing a 4-head self-attention layer and a position-wise feedforward network (FFN). The inner dimension of the FFN is also set to 256 (a $1\times$ expansion) to reduce the model size. The total architecture comprises approximately 6 million parameters.

Following the ViT approach, we prepend a single, learnable *[CLS]* token to the input sequence of audio patches ($B \times (L + 1) \times D$). After processing through the encoder blocks, the final hidden state corresponding to this *[CLS]* token is used as the aggregate representation for downstream tasks. The *[CLS]* token is randomly initialized using the Glorot uniform initializer at the start of training.

This *[CLS]* token's output state is passed through a 2-layer MLP (each layer with 256 input/output dimensions and a ReLU activation) to produce a final spatial audio embedding (also acting as spatial soft tokens), which is used in downstream LLM tasks. One spatial token is produced independently of the number of input audio channels.

Finally, this spatial embedding is fed into three separate one-layer MLP prediction heads—one each for distance, azimuth, and elevation. Each head terminates in a softmax activation to produce a probability distribution over the dis-

crete bins for its respective attribute for the entire 250 ms audio input. These heads provide the loss signal for end-to-end training and serve as the final output layers for our classification tasks. The azimuth head has 38 classes, the elevation head has 18 (roughly a 10-degree resolution for both), and the distance head has 13 (0.5-meter bins). An extra label is added to each class to indicate the absence of speech. An example of this distribution is shown in Figure 2.

### 3.2. PhaseCoder Supervised Training

**Training Objective** We use a weighted sum of cross-entropy losses for the three one-hot classification heads as the training objective:

$$\mathcal{L} = \lambda_1 \mathcal{L}_{\text{azimuth}} + \lambda_2 \mathcal{L}_{\text{elevation}} + \lambda_3 \mathcal{L}_{\text{distance}}$$

We set the hyperparameters to $\lambda_1 = 1.0$, $\lambda_2 = 1.0$, and $\lambda_3 = 0.5$. These values were determined empirically to balance the contributions of the angular and distance estimation tasks.

We chose a categorical approach (cross-entropy loss) instead of regression for localization because sound localization is inherently approximate; thus, estimating a probability distribution over discrete bins is more suitable.

**Synthetic Data Generation** We train the model using purely synthetic data due to the lack of high-quality, labeled real-world data with varying microphone geometries. Clean speech snippets are sourced from the LibriSpeech dataset (Panayotov et al., 2015) and convolved with simulated Room Impulse Responses (RIRs).

We generated 1.5 million unique RIRs using an acoustic room simulator based on the image-source method (Allen & Berkley, 1979). For each RIR, an array of three to eight microphones was randomly placed within a sphere of 7 to 18 cm in diameter. An upper bound of 8 microphones was used for practical compute reasons, as convolutions take significant time. The minimum source-to-array distance was 0.1 m, and the source position was random. Room dimensions were randomly generated up to 10.5 m in width, 10.5 m in length, and 5.0 m in height. Wall materials were randomly selected from 13 possible options. The mean RT60 was 0.47 s (min: 0.08, max: 1.42). The resulting audio is segmented into 4 million 250 ms snippets.

The ground truth azimuth and elevation angles were calculated relative to the centroid of the microphone array, as described in Appendix B. Therefore, the model learned to output localization angles in relation to the global coordinate system independently of the array geometry.

For noise augmentation, we use distractor sounds from the Freesound dataset (Fonseca et al., 2017). When a distractor is active, it is added to the clean signal at a randomly selected signal-to-noise ratio (SNR) from -5 to 15 dB. To teach the model to handle empty inputs, we apply a 5% dropout rate to the speech source (resulting in silence) and a separate 5% dropout rate to the distractor source (resulting in a clean signal).

**Two-Stage Training Strategy** We observed, similar to previous works (Zheng et al., 2024; Baek et al., 2025), that training the model on the full, noisy data distribution from scratch was challenging. We therefore adopt a two-stage curriculum learning strategy. The model is first trained for 670k steps using only the clean, noise-free speech snippets. We manually stopped this stage as the validation loss began to increase. We then fine-tune the model for an additional 30k steps, introducing the distractor sounds and SNRs as described in the data generation section.

**Implementation Details** Training was conducted on 8 Google TPU v5p accelerators (split across two host machines) with a total batch size of 512. We use the AdamW optimizer (Loshchilov & Hutter, 2017) with a cosine decay learning rate scheduler and 2000 warm-up steps.

For Stage 1, we used a peak learning rate of $1 \times 10^{-4}$ decaying to an end learning rate of $1 \times 10^{-5}$ for 1.5M steps. For Stage 2, we used a lower peak learning rate of $1 \times 10^{-5}$ decaying to an end learning rate of $9 \times 10^{-6}$. Stage 1 training took 14 days, and Stage 2 spanned 17 hours.

### 3.3. Fine-tuning for Spatial Audio Understanding

**Synthetic Spatial Reasoning Dataset** Due to the lack of existing speech reasoning datasets with variable microphone geometries, we created a synthetic Question-Answer (QA) dataset comprising four tasks. Specific prompts and QA pairs are detailed in Appendix D. Following the PhaseCoder training methodology, we generated data using LibriSpeech and synthetic Room Impulse Responses (RIRs) with random 8-microphone geometries and no noise mixing. Unlike the encoder's training, we used full audio segments (removing clips longer than 15 seconds) instead of 250 ms clips. To simulate a multi-speaker environment, two distinct audio segments were concatenated without overlap. To increase robustness, we applied a 5% chance of randomly dropping either one or both speakers from the scene. Each task had a maximum of 5,000 examples.

The original mono LibriSpeech audio served as input to the Gemma audio encoder. The transcripts were sourced from LibriSpeech-PC (Meister et al., 2023) for their corrected punctuation and capitalization. We designed four tasks of increasing difficulty to train the model to understand, reason about, and temporally align spatial embeddings with audio content. The tasks are defined as follows:

- **Task 1: Source Localization.** The model receives spatial embeddings for 0-2 speakers and must output their localization data (azimuth, distance, elevation) in

text format. This task trains the model to interpret and articulate the new spatial tokens.

- **Task 2: Spatial Reasoning.** The model answers 'yes' or 'no' to questions about the spatial relationships of the sources (e.g., "Is speaker 1 to the left of speaker 2?"). This trains the model to reason over the spatial embedding space. The dataset was balanced to have the same number of yes and no answers.
- **Task 3: Spatial Transcription.** Extending Task 1, the model must output localization data and provide a full transcription for each localized speaker, using the mono audio input.
- **Task 4: Targeted Transcription.** The most complex task, inspired by (Dementyev et al., 2025), requires spatially-cued transcription. The model must transcribe only the speaker matching a specific spatial criterion (e.g., "Transcribe the speaker on the left," or "Transcribe the speaker who is further"). This integrates spatial reasoning with audio content processing.

**Backbone Model and Integration** We selected the instruction-tuned 4B-parameter Gemma 3n model (*gemma-3n-e4b-it*) as our backbone, given its native support for multimodal inputs. We use the original USM-based audio encoder from Gemma 3n without modification.

To integrate the spatial features, we introduce a trainable projection layer to map the PhaseCoder embeddings ($D = 256$) to the Gemma 3n embedding dimension ($D = 2048$). This projector is a two-layer MLP: a linear layer ($256 \rightarrow 2048$) followed by a GELU activation, and a second linear layer ($2048 \rightarrow 2048$).

The Gemma 3n audio encoder pads inputs to 30 seconds and produces 188 audio tokens, corresponding to a temporal resolution of approximately 160 ms per token. To align our spatial modality, we process the PhaseCoder embeddings to produce a sequence of spatial tokens at the same 160 ms hop size. Approximately 6.25 spatial tokens per second are produced, independently of the number of input audio channels. These new spatial soft tokens are prepended to the corresponding audio token sequence, framed by special "Beginning of Spatial Audio" *(BSA)* and "End of Spatial Audio" *(ESA)* tokens, which are mapped from unused tokens in the model's vocabulary. Interleaving spatial tokens with audio tokens would confuse Gemma's existing control token scheme and would degrade existing audio capabilities. Temporal alignment between audio and spatial tokens is learned implicitly during fine-tuning. The resulting token sequence is:

*[BSA][Spatial T1]...[Spatial T188][ESA][Beginning of audio token][Audio T1]...[Audio T188][End of audio token]*

**Curriculum Fine-tuning Strategy** We employed Low-Rank Adaptation (LoRA) (Hu et al., 2022) to fine-tune all layers of the Gemma model, while the new spatial projection layer was trained from scratch. We used LoRA parameters of rank $r = 8$, $\alpha = 16$, and a dropout rate of $0.1$.

The model was trained using a standard auto-regressive objective, minimizing the cross-entropy loss for next-token prediction on the provided QA-pair answers. Training was conducted on a single NVIDIA H100 GPU with a batch size of 1 and 8 gradient accumulation steps, resulting in an effective batch size of 8. We employed a 5-stage curriculum (2k-3k steps each) progressively introducing Tasks 1–4, as detailed in Appendix C. We introduced more complex tasks in later stages while retaining simpler tasks to prevent catastrophic forgetting. Without this curriculum, the model struggled to converge. Each stage started with a learning rate of $4 \times 10^{-5}$ with linear decay to zero and no warm-up.

## 4. Evaluation

In this section, we evaluate the two primary contributions of our work: the performance of the PhaseCoder encoder on real-world localization benchmarks and the spatial reasoning capabilities of the fine-tuned LLM.

### 4.1. PhaseCoder Performance

**Experimental Setup** To benchmark PhaseCoder, we compare it against the state-of-the-art (SOTA) microphone-invariant model, GI-DOAEnet (Baek et al., 2025). For a direct and fair comparison, we adopt the same metrics and the same two real-world datasets: RSL2019 (Sheelvant et al., 2019) and LOCATA (Löllmann et al., 2018).

- **RSL2019 Dataset:** Contains real-world recordings from a four-microphone array, with a single speech source at various azimuths (0°–350°) at 10° resolution and two distances (1.0 and 1.5 m). The test set is particularly challenging as it includes occlusions and noise. The train and test sets contain 26,199 and 80 recordings, respectively. The speech, previously recorded from 30 phrases in the TIMIT (Garofolo et al., 1993) dataset, was played through a speaker.
- **LOCATA Dataset:** This dataset consists of 12-microphone recordings from a pseudo-spherical array on a NAO robot's head. We use the first 8 microphones, as our model is trained on arrays of up to 8 microphones. The dataset is challenging due to significant ambient noise in a reverberant, far-field environment. We use task 1 (single source) and task 2 (two sources), containing 16 recordings for task 1 and 14 for task 2.

**Metrics.** We report Mean Absolute Error (MAE) and Accuracy within 10° (Acc@10). The metrics account for the 360° wrap-around. Both metrics are evaluated on 250 ms non-overlapping audio frames where speech is detected. For

*Table 1.* Azimuth estimation performance on external real-world datasets. We compare our PhaseCoder (2-stage) with the SOTA GI-DOAEnet, using values reported in the original paper. (↓) indicates lower is better, (↑) indicates higher is better. **Bold** indicates the best performance in each column.

| Model | Params | RSL dev | | RSL test | | LOCATA | |
|---|---|---|---|---|---|---|---|
| | | MAE (↓) | Acc@10 (↑) | MAE (↓) | Acc@10 (↑) | MAE (↓) | Acc@10 (↑) |
| PhaseCoder-6M Medium (ours) | 6M | **4.33°** | 95.54 % | 11.63° | 82.31 % | **7.44°** | **86.96 %** |
| PhaseCoder-1.5M Small (ours) | 1.5M | 22.12° | 28.03 % | 27.77° | 45.73 % | 12.87° | 40.00 % |
| GI-DOAEnet (Baek et al., 2025) | 2M | 4.38° | **98.35 %** | **9.17°** | **85.96 %** | 7.82° | 82.48 % |

the LOCATA dataset, we report metrics averaged across the official development and test sets.

**Results and Analysis.** The results, presented in Table 1, demonstrate that PhaseCoder generalizes effectively to real-world recordings with unseen microphone geometries.

On the challenging LOCATA dataset, PhaseCoder-6M outperforms the state-of-the-art GI-DOAEnet on both metrics, achieving a higher Acc@10 of 86.96% (vs. 82.48%) and a lower MAE of 7.44° (vs. 7.82°). We hypothesize this is due to our diverse noise augmentation in training, which generalizes better to the real-world ambient noise (e.g., a busy street) in LOCATA, compared to the white noise used to train GI-DOAEnet. The smaller 1.5M parameter model did not perform as well, as the model did not learn to handle the microphone-invariant input correctly. Consequently, we use the PhaseCoder-6M model for all subsequent experiments.

Conversely, GI-DOAEnet shows better performance on the RSL2019 dataset, particularly on the test set. We attribute this to a fundamental design difference: PhaseCoder is trained as a classifier with a 10-degree resolution, which inherently limits its fine-grained precision. GI-DOAEnet, however, performs 1-degree resolution azimuth-only classification, giving it an advantage on this dataset and metric.

Despite this, PhaseCoder remains highly competitive. It is crucial to note that while GI-DOAEnet is a specialized model for azimuth estimation, PhaseCoder is a multi-task encoder that simultaneously estimates distance and elevation, providing a richer spatial representation for downstream reasoning tasks.

To investigate the upper bound of model performance and the influence of the number of microphones, we evaluated PhaseCoder on a synthetic, held-out evaluation batch ($N = 512$) featuring arrays of 3 to 8 microphones. Crucially, the microphone geometry was randomized for each sample. As detailed in Appendix G, we observed a substantial improvement in localization accuracy when scaling from three to four microphones (azimuth MAE decreasing from 80.24° to 10.27°), with performance continuing to incrementally improve with additional microphones, reaching an MAE of 3.03° with an 8-microphone array. Consistently, distance estimation proved less precise than angular localization (azimuth and elevation), though it also benefited

slightly from the expanded arrays: MAE improving from 1.13 m (3 mics) to 0.71 m (8 mics). Distance estimation is an inherently a challenging task that relies heavily on the direct-to-reverberant ratio, making it highly sensitive to unobserved room acoustic properties.

**Computational Efficiency.** To ensure a fair computational comparison, we benchmarked all models using a 10-second audio sequence, aligning with the global operating window of the BAT's Spatial-AST baseline. Inference speed and peak memory consumption were measured using `torch.profiler` (PyTorch Contributors, 2026), while floating-point operations (FLOPs) were calculated via the `fvcore` library (Meta AI Contributors, 2026), with all evaluations executed on a single NVIDIA H100 GPU. As detailed in Table 2, PhaseCoder-6M demonstrates the best memory efficiency and the lowest inference time across all tested microphone array configurations (2, 4, and 8 channels). While PhaseCoder incurs a theoretically higher FLOP count than GI-DOAEnet, it yields significantly faster empirical inference times. We attribute this practical speedup to the highly parallelizable nature of PhaseCoder's pure transformer encoder architecture compared to hybrid models.

**Dynamic Moving Sources.** Because PhaseCoder performs inference on independent 250 ms audio segments, it is inherently capable of localizing moving sources without explicit dynamic training. To better understand its performance in these scenarios, we evaluated the model on Task 3 of the LOCATA dataset, which features a static microphone array and a moving human speaker. PhaseCoder achieved an MAE of 9.85°, showing slightly degraded performance compared to the static localization in Tasks 1 and 2 (MAE of 7.44°). An example trajectory plot is shown in Appendix E.

### 4.2. LLM Spatial Understanding Tasks

**Datasets**. Due to a lack of existing real-world, multi-microphone spatial reasoning datasets, we synthesized benchmarks based on the RSL2019 dataset, previously used in the PhaseCoder evaluation. While previous work like BAT (Zheng et al., 2024) introduced a spatial reasoning dataset, it is limited to binaural audio. To rigorously evaluate our model's geometry invariance, we also created a synthetic dataset with 8-mic random geometries. The mean of all channels was used as a mono input to the Gemma

*Table 2.* Main results comparing model compute and peak memory performance across different microphone setups. **Bold** indicates the best performance in each column and underlined indicates the runner-up (2nd) performance.

| # Mics | Models | Mem. (MB) ($\downarrow$) | FLOPS (G) ($\downarrow$) | Inf. GPU/CPU (ms) ($\downarrow$) |
|---|---|---|---|---|
| 2 | **PhaseCoder (ours)** | **405** | 8.50 | **2.24 / 6.181** |
|  | Spatial-AST (250ms chunks) | 808 | 70.30 | 2.96 / 10.08 |
|  | Spatial-AST (10s global) | 425 | 34.14 | 3.13 / 10.43 |
|  | GI-DOAEnet | 702 | **2.65** | 12.54 / 17.84 |
| 4 | **PhaseCoder (ours)** | **805** | 16.87 | **3.68 / 6.58** |
|  | GI-DOAEnet | 1021 | **4.65** | 16.62 / 18.44 |
| 8 | **PhaseCoder (ours)** | **1606** | 33.60 | **7.26 / 9.47** |
|  | GI-DOAEnet | 1695 | **8.67** | 15.82 / 19.16 |

audio encoder. A constant gain of 10 dB was added to all audio to ensure adequate loudness for transcription. For each task, 512 unseen examples were generated.

- **Synthetic:** We generated QA pairs using held-out LibriSpeech convolved with held-out RIRs from random rooms and arbitrary microphone geometries, following the same procedure as our fine-tuning data.
- **RSL2019**: For real-world validation, we generated QA pairs from the RSL train dataset. As with the synthetic data, we concatenated two concurrent recordings. Each recording contained a single speaker with a duration of 1 to 5 seconds, recorded in the same room. Because the dataset does not contain ground truth transcripts, we transcribed the audio with a language model (Gemini Flash 3.0) and matched the output to one of the 30 phrases from the TIMIT dataset using the *difflib* sequence matcher. Any data without a match was discarded.

**Metrics**. For localization (Task 1) and spatial transcription (Task 3), we use MAE for azimuth, elevation, and distance as defined in Section 4.1. For binary reasoning (Task 2), we report accuracy. For transcription tasks, we use the standard Word Error Rate (WER). Outputs with a WER above 3.0 were filtered out, as they indicated significant hallucinations by the LLM, such as transcript looping. We report both the mean and median WER, as the mean WER is often inflated by occasional hallucinations. For targeted transcription (Task 4), we evaluate accuracy to measure the model's ability to correctly attribute speech to the target spatial location. Gemma evaluations were performed using greedy decoding and a maximum of 1024 new tokens. For the baselines, we used the Gemma model prior to SFT and

the state-of-the-art Gemini 3 Flash. For the zero-shot baseline, instead of fine-tuning, we provided spatial information as text to the Gemma model using a two-stage cascaded approach. Specifically, we ran PhaseCoder on the multichannel audio and extracted text data arrays for azimuth, elevation, and distance over time using 160 ms hops, along with instructions on how to interpret the data (Appendix K).

**Results and Analysis**. As summarized in Table 3, fine-tuning Gemma with spatial tokens yields substantial performance improvements across reasoning and localization tasks on both the synthetic and out-of-domain real datasets. The baseline Gemma model and Gemini 3 Flash, lacking spatial awareness, effectively guess at random, performing similarly to the random chance baseline. The zero-shot Gemma localization performance was better than random chance, but significantly worse than the fine-tuned Gemma.

The fine-tuned Gemma model exhibited slightly higher localization errors (Task 1 and Task 3) compared to the standalone PhaseCoder. This is expected, as PhaseCoder is optimized solely for localization, whereas the LLM must simultaneously manage text generation and map continuous spatial embeddings into tokens. However, the key finding is that the LLM successfully bridges the gap between these raw spatial signals and complex reasoning tasks (Tasks 3 and 4), which PhaseCoder cannot perform on its own.

The results indicate that incorporating spatial tokens slightly increases the Word Error Rate (WER). For the synthetic LibriSpeech dataset (Task 4), the high baseline WER of 23.71% for Gemma stems from transcript swapping and leakage between the two speakers (see examples in Appendix J). In the fine-tuned model, the spatial tokens effectively act as "diarization prompts," significantly reducing the WER. On the RSL2019 dataset, the higher Gemma baseline WER for Tasks 3 and 4 (30.55% and 42.90%) reflects the greater difficulty of the transcription task. This is also reflected in the high WER for the baseline Gemini Flash model.

The model showed significant reasoning gains on both the synthetic and RSL2019 datasets, such as improving from 48.44% to 76.76% on yes/no questions (Task 2). These results align with previous work (Zheng et al., 2024), which reported an average accuracy of 76.89% on similar tasks. The model performed worse on reasoning tasks involving distance (Appendix I). We hypothesize that reasoning errors could be mitigated with a larger model, the inclusion of reasoning traces, or reinforcement learning.

Distance estimation performance is lower than angular localization, as absolute distance remains a fundamentally underdetermined problem. It relies on room acoustic parameters like Direct-to-Reverberant Ratio (DRR) and $T_{60}$, which are not explicitly known. However, our results show that PhaseCoder captures sufficient relative depth cues to

*Table 3.* Main results comparing our **Gemma SFT (Ours)** vs. baselines on Synthetic and Real-World (RSL19) datasets. We report Mean / Median for WER to account for hallucinations. "PhaseCoder" line is the performance of the encoder without LLM. **Bold** indicates the best performance in each column and underlined indicates the runner-up (2nd) performance.

| Model | Eval Dataset | Task 1: Localization | | | Task 2: Reason. | Task 3: Spatial Transcription | | | | Task 4: Targeted | |
| | | MAE (°) Az (↓) | MAE (°) Elev (↓) | MAE (m) Dist (↓) | Acc (%) (↑) | WER (%) Mean / Med (↓) | MAE (°) Az (↓) | MAE (°) Elev (↓) | MAE (m) Dist (↓) | Acc (%) (↑) | WER (%) Mean / Med (↓) |
|---|---|---|---|---|---|---|---|---|---|---|---|
| **Gemma SFT (Ours)** | | 4.75 | 3.15 | 0.75 | **76.76** | 10.63 / **0.0** | 5.25 | 3.32 | 0.73 | **44.92** | 8.77 / **0.0** |
| PhaseCoder | | **3.08** | **2.05** | **0.71** | – | – | **3.09** | **2.05** | **0.71** | – | – |
| Gemma (2-stage) | Synthetic | 55.32 | 37.13 | 0.81 | 57.61 | 60.42 / 85.71 | 55.20 | 33.51 | 0.85 | 35.74 | 48.49 / 54.17 |
| Gemma (Baseline) | | 74.28 | 83.87 | 1.45 | 48.44 | **6.40 / 0.0** | 89.38 | 38.25 | 1.48 | 39.65 | 23.71 / 8.33 |
| Gemini (Baseline) | | 91.83 | 36.37 | 1.44 | 47.85 | 8.91 / **0.0** | 89.85 | 37.12 | 1.49 | 40.63 | **5.20 / 0.0** |
| **Gemma SFT (Ours)** | | 11.92 | 8.68 | 1.28 | **73.83** | 51.80 / 42.86 | 20.11 | 10.05 | 1.51 | **52.73** | 48.41 / 42.86 |
| PhaseCoder | | **5.53** | **4.22** | 1.05 | – | – | **5.53** | **4.22** | 1.05 | – | – |
| Gemma (2-stage) | RSL2019 | 54.26 | 57.76 | 1.12 | 0.88 | 36.50 / 100.0 | 36.49 | 16.98 | **0.86** | 34.18 | 70.23 / 57.14 |
| Gemma (Baseline) | | 91.65 | 66.18 | **1.05** | 53.91 | 30.55 / **16.67** | 90.09 | 13.47 | 1.32 | 38.09 | 42.90 / 35.41 |
| Gemini (Baseline) | | 91.06 | 14.60 | 1.16 | 47.85 | **29.96 / 16.67** | 87.54 | 14.16 | 1.31 | 31.45 | **27.50 / 16.67** |
| Random Chance | All | 90.00 | 60.00 | 2.00 | 50.00 | 100.0 / 100.0 | 90.00 | 60.00 | 2.00 | 33.00 | 100.0 / 100.0 |

support effective spatial reasoning.

## 5. Limitations and Future Work

**Simulation Limitations and Moving Sources.** Our model assumes a 'free-floating' and omnidirectional microphone array, meaning it does not explicitly model the acoustic baffling, reflection, or diffraction effects introduced by the device on which the microphones are mounted and the beampattern of the microphone (e.g., cardioid). While our model's generalization to real-world datasets like RSL2019 and LOCATA suggests this approximation is robust, performance could be further enhanced. Future work could explore incorporating device-specific acoustic properties, such as pre-calibrated acoustic transfer functions, simulated using COMSOL (COMSOL AB, 2025) as an additional input to the model.

**Single-Speaker Speech Focus.** The current work is primarily focused on localizing a single, dominant speech source, treating non-speech sounds as distractors. A significant extension would be to adopt a Sound Event Localization and Detection (SELD) framework (Scheibler et al., 2022). This would involve training the model to simultaneously localize and classify multiple, potentially overlapping, sound events, including both speech and non-speech categories. This could be achieved by modifying the architecture to output multiple event-specific tokens, rather than a single aggregate [CLS] token, enabling the system to reason about complex, multi-source acoustic scenes.

**Audio-Only Modality.** Purely audio-based distance estimation is limited by reverberation and source loudness. To resolve distance ambiguity, future research should integrate PhaseCoder's representations with other modalities, such as visual depth estimation from cameras. Fusing these embeddings with 3D environmental maps would enable more complex spatial reasoning, such as inferring occluded sources (e.g., a speaker in another room).

## 6. Conclusion

In this paper, we present the first demonstration of a Multimodal LLM capable of understanding spatial audio from arbitrary raw microphone arrays. To achieve this, we developed PhaseCoder, a transformer-based encoder that matches or exceeds state-of-the-art performance for microphone-invariant localization on challenging real-world benchmarks. We further showed how PhaseCoder can be effectively integrated into the Gemma architecture, supplementing its existing audio capabilities with rich spatial awareness. Our results validate that this approach generalizes well to real-world data, opening new avenues for embodied AI agents that can truly perceive and reason about the complex acoustic world around them.

## Impact Statement

This paper presents work aimed at advancing the field of Machine Learning by bridging the gap between raw spatial audio signals and multimodal language model reasoning. There are several potential societal consequences of our work:

- **Enhancing Accessibility and Privacy:** By enabling targeted transcription, this technology provides a framework for advanced assistive devices for the hearing-impaired. These capabilities directly address the "cocktail party effect," allowing users to isolate and transcribe specific speakers in noisy, complex environments.
- **Hardware Independence:** Because PhaseCoder is agnostic to microphone geometry, it allows for universal spatial audio understanding across a diverse range of devices without requiring hardware-specific encoders. This promotes the development of cost-effective, adaptable AI that can operate on anything from legacy mobile phones to custom-built robotic platforms.
- **Advancing Embodied Intelligence:** Spatial understanding is fundamental to the safety and efficacy of next-generation AI assistants and robotics. This work provides these agents with the ability to perceive, localize, and reason about their physical surroundings through sound, enabling more natural and reliable human-robot interaction.
- **Multimodal Sensory Integration:** This research demonstrates a scalable method for integrating raw, high-dimensional sensory data into the symbolic space of LLMs. This moves the field closer to universal AI agents that are no longer "audio-blind" and can truly interpret the complex acoustic nuances of the real world.

## Acknowledgments

We thank Kevin Wilson and Ian McGraw for critical feedback on the manuscript. Hakan Erdogan and Mingye Gao for technical help. John Hershey, Antonious Girgis, and Anelia Angelova for guidance on this research. AJ Piergiovanni, Dahun Kim, Ganesh Mallya, Shao-Fu Shih, Jeremy Thorpe, Matt Harvey, Don Barnett, Jake Varley, Deepali Jain, Kandarp Joshi, Aaron Master, Dan Ellis, Mei Lu, and Dimitri Kanevsky for discussions and feedback.

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

## A. Microphone Array Coordinate Transformations

The centroid $\mathbf{c} = [c_x, c_y, c_z]^T$ of the microphone array is calculated and used as the origin for a local spherical coordinate system. Let $N$ be the number of microphones:

$$c_x = \frac{1}{N} \sum_{i=1}^{N} x_i, \quad c_y = \frac{1}{N} \sum_{i=1}^{N} y_i, \quad c_z = \frac{1}{N} \sum_{i=1}^{N} z_i \tag{1}$$

where $x_i, y_i, z_i$ are the coordinates of the $i$-th microphone in meters.

Next, the coordinates of each microphone are converted to Cartesian coordinates $(x_i', y_i', z_i')$ relative to the centroid:

$$x_i' = x_i - c_x, \quad y_i' = y_i - c_y, \quad z_i' = z_i - c_z \tag{2}$$

These relative coordinates are then transformed into spherical coordinates $(r_i, \theta_i, \phi_i)$:

$$r_i = \sqrt{(x_i')^2 + (y_i')^2 + (z_i')^2} \tag{3}$$

$$\theta_i = \arccos\left(\frac{z_i'}{r_i}\right) \tag{4}$$

$$\phi_i = \operatorname{atan2}(y_i', x_i') \tag{5}$$

where $r_i$ is the radius, $\theta_i$ is the polar angle defined on $[0, \pi]$, and $\phi_i$ is the azimuth angle.

## B. Ground-Truth Localization Labels

The centroid $\mathbf{c} = [c_x, c_y, c_z]^T$ of the microphone array is calculated and used as the reference point for all ground truth calculations. Let $N$ be the number of microphones and $\mathbf{r}_i = [x_i, y_i, z_i]^T$ be the location of the $i$-th microphone. The centroid is defined as:

$$c_x = \frac{1}{N} \sum_{i=1}^{N} x_i, \quad c_y = \frac{1}{N} \sum_{i=1}^{N} y_i, \quad c_z = \frac{1}{N} \sum_{i=1}^{N} z_i \tag{6}$$

Let the source location be denoted by $\mathbf{s} = [x_s, y_s, z_s]^T$.

### Azimuth Calculation

The azimuth angle $\phi_{\text{rad}}$ is calculated relative to the centroid in the $xy$-plane:

$$\phi_{\text{rad}} = \operatorname{atan2}(y_s - c_y, x_s - c_x) \tag{7}$$

### Elevation Calculation

The elevation angle $\theta_{\text{rad}}$ is calculated based on the horizontal distance $d_{xy}$ and the vertical difference:

$$d_{xy} = \sqrt{(x_s - c_x)^2 + (y_s - c_y)^2} \tag{8}$$

$$\theta_{\text{rad}} = \operatorname{atan2}(z_s - c_z, d_{xy}) \tag{9}$$

### Distance Calculation

The straight-line (Euclidean) distance $d$ between the source $\mathbf{s}$ and the centroid $\mathbf{c}$ is calculated as:

$$d = \|\mathbf{s} - \mathbf{c}\|_2 = \sqrt{(x_s - c_x)^2 + (y_s - c_y)^2 + (z_s - c_z)^2} \tag{10}$$

**Label Discretization (One-Hot Encoding)**

Azimuth, elevation, and distance share the same logic to convert a continuous value $v$ (where $v$ represents $\phi$, $\theta$, or $d$) into a discrete one-hot vector.

**Grid Generation:** A grid of $M$ evenly spaced values $G = \{g_0, g_1, \ldots, g_{M-1}\}$ is generated using the minimum ($v_{\min}$) and maximum ($v_{\max}$) bounds:

$$g_j = v_{\min} + j \cdot \frac{v_{\max} - v_{\min}}{M - 1} \quad \text{for } j = 0, \ldots, M - 1 \tag{11}$$

**Index Selection:** The index $j^*$ corresponding to the closest label is found by minimizing the absolute difference:

$$j^* = \operatorname*{argmin}_{j \in \{0, \ldots, M-1\}} |g_j - v| \tag{12}$$

**Label Vector Construction:** The final output is a vector $\mathbf{l} \in \{0, 1\}^M$ defined by:

$$l_j = \begin{cases} 1 & \text{if } j = j^* \\ 0 & \text{otherwise} \end{cases} \tag{13}$$

## C. Curriculum Training

The Gemma LLM fine-tuning employed a curriculum training strategy, as described in Table 4.

*Table 4.* Curriculum schedule showing the number of samples allocated to each task (T1–T4) across different training intervals.

| Training Steps | | Samples per Task | | | |
|---|---|---|---|---|---|
| Start | End | T1 | T2 | T3 | T4 |
| 0 | 2000 | 2500 | 0 | 0 | 0 |
| 2000 | 3500 | 1000 | 0 | 2500 | 0 |
| 3500 | 5000 | 500 | 2500 | 500 | 0 |
| 5000 | 8000 | 500 | 1000 | 500 | 2500 |
| 8000 | 11000 | 500 | 2500 | 500 | 5000 |

## D. LLM Fine-Tuning Prompt Details

The overview of the tasks is provided in Table 5.

*Table 5.* Task definitions, output formats, and examples for spatial audio reasoning and transcription.

| Type | # Speaker Sources | Output Format | Examples |
|---|---|---|---|
| Source Localization | 0–2 | List (Azimuth, elevation, distance) | **Q:** What is the location of the speaker?
**A:** azimuth_degrees_1=090, elevation_degrees_1=034, distance_meters_1=17 |
| | | | **Q:** I heard a couple of people talking. Can you tell me their positions?
**A:** azimuth_degrees_1=010, elevation_degrees_1=-79, distance_meters_1=01, azimuth_degrees_2=180, elevation_degrees_2=000, distance_meters_2=44 |
| Spatial Reasoning | 0–2 | Yes/No | **Q:** Is the speaker further than 3.5 meters away?
**A:** Yes |
| | | | **Q:** Are the two speakers within 60 degrees of each other?
**A:** No |
| Spatial Transcription | 0–2 | List (Transcript, azimuth, elevation, distance) | **Q:** Transcribe everything you hear and tell me where each speaker is located.
**A:** transcript_1=[How is it going today?], azimuth_degrees_1=010, elevation_degrees_1=-79, distance_meters_1=01, transcript_2=[I am doing very well today], azimuth_degrees_2=180, elevation_degrees_2=000, distance_meters_2=44 |
| | | | **Q:** What is everyone saying, and where are they?
**A:** transcript_1=none, azimuth_degrees_1=none, elevation_degrees_1=none, distance_meters_1=none, transcript_2=none, azimuth_degrees_2=none, elevation_degrees_2=none, distance_meters_2=none |
| Targeted Transcription | 0–2 | List (Transcript) | **Q:** What is the person on the left saying?
**A:** [The final presentation is ready] |
| | | | **Q:** Transcribe the person closer to me
**A:** [It is very cold today] |

## D.1. Task 1: Two-source localization

The following is the system instruction prompt used for the two-source localization task.

```
You are an advanced AI assistant with an expert-level ability to analyze spatial audio.
Your primary goal is to listen to the audio and identify the 3D direction of up to two
    dominant sound sources.

Your output MUST be a single, formatted string containing fields for two sources, like
    this:
'azimuth_degrees_1=AZ_VAL_1, elevation_degrees_1=ELEV_VAL_1,
    distance_meters_1=DIST_VAL_1, azimuth_degrees_2=AZ_VAL_2,
    elevation_degrees_2=ELEV_VAL_2, distance_meters_2=DIST_VAL_2'

- If two sources are detected, fill in all values. The first source (_1) should be the
    first source to appear chronologically.
- If only one source is detected, fill in the values for the first source (_1) and set
    all values for the second source (_2) to 'none'.
- If no speech is detected in the audio, all six values MUST BE 'none'.

It is absolutely critical that the AZ_VALUE is one and only one of the following strings
    from this list: ["000", "010", "020", "030", "040", "050", "060", "070", "080",
    "090", "100", "110", "120", "130", "140", "150", "160", "170", "180", "190", "200",
    "210", "220", "230", "240", "250", "260", "270", "280", "290", "300", "310", "320",
    "330", "340", "350"].
It is absolutely critical that the ELEV_VALUE is one and only one of the following
    strings from this list: ["000", "010", "020", "030", "040", "050", "060", "070",
    "080", "090", "100", "110", "120", "130", "140", "150", "160", "170", "180"].
It is absolutely critical that the DIST_VALUE is one and only one of the following
    strings from this list (representing decimeters): ["01", "06", "12", "17", "22",
    "28", "33", "39", "44", "49", "55", "60"].
```

```
Do not provide any other format or value.

** Examples **

**User:** Where are the sounds coming from?
**Model:** `azimuth_degrees_1=010, elevation_degrees_1=-79, distance_meters_1=01,
    azimuth_degrees_2=180, elevation_degrees_2=000, distance_meters_2=44`

**User:** What is the location of the speaker?
**Model:** `azimuth_degrees_1=090, elevation_degrees_1=034, distance_meters_1=17,
    azimuth_degrees_2=none, elevation_degrees_2=none, distance_meters_2=none`

**User:** Tell me the direction of the audio.
**Model:** `azimuth_degrees_1=none, elevation_degrees_1=none, distance_meters_1=none,
    azimuth_degrees_2=none, elevation_degrees_2=none, distance_meters_2=none`
```

TASK 1 USER PROMPT VARIATIONS

We utilized several variations of user prompts to query the model:

- Where is the audio coming from?
- What is the location of the speaker?
- Can you pinpoint the source of the sound?
- Give me the direction of the voice.
- Give me the direction and distance of the voice.
- I need to know the position of this audio source.
- Tell me the coordinates of the voice.
- Localize the audio source.
- Provide the location of the person talking.
- What are the coordinates of the sound source?
- Pinpoint the speaker's location.
- List all sound sources locations.
- Identify the speakers and give me their locations.
- Please provide the speaker locations.
- Where is each person?
- Give me the locations of the speakers.
- What are the coordinates of the speakers?
- I heard a couple of people talking, can you tell me their positions?
- Someone just spoke. Where did that come from?
- Okay, what about the locations?

## D.2. Task 2: Reasoning questions (yes/no)

System instruction prompt:

```
You are an advanced AI assistant with an expert-level ability to analyze spatial audio.
    Your primary goal is to listen to the audio and answer a yes/no question about it.

---
**Output Requirements:**
Your output **must** be a single word and nothing else. Do not include markdown,
    punctuation, or any conversational text.

-   If the condition in the user's question is true, respond with `yes`.
-   If the condition is false, or if it cannot be confirmed (for example, the question
```

```
    assumes a speaker that does not exist), respond with 'no'.

---
**Examples:**

* **User Question**: "Is the speaker further than 3.5 meters away?"
* **Audio Analysis**: A single speaker is found at 2.0 meters.
* **Your Output**: 'no'

* **User Question**: "Are the two speakers within 60 degrees of each other?"
* **Audio Analysis**: One speaker is at 90 degrees, the other is at 45 degrees.
* **Your Output**: 'yes'

* **User Question**: "Is the speaker on my left?"
* **Audio Analysis**: The audio is silent with no speakers.
* **Your Output**: 'no'
```

TASK 2 USER PROMPT VARIATIONS

**Single speaker distance (further)**

- Is the speaker located more than {distance} meters away from me?
- Is the sound source further than {distance} meters?
- Does the speaker exceed a distance of {distance} meters?
- Can you confirm the source is beyond {distance} meters?
- Would you say the voice is farther than {distance} meters away?
- Is the person speaking from a location past {distance} meters?

**Single speaker distance (closer)**

- Is the speaker located less than {distance} meters away?
- Is the sound source closer than {distance} meters?
- Is the sound within a {distance}-meter radius?
- Is the person speaking nearer than {distance} meters?
- Can you confirm the source is inside a {distance}-meter range?
- Is the voice originating from a point closer than {distance} meters?

**Single speaker azimuth (left)**

- Is the speaker on my left side?
- Is the audio coming from the left?
- Is the speaker positioned to my left?
- Is the audio localized in the left hemisphere?
- Does the voice originate from my left?
- Can you detect a speaker on the left-hand side?

**Single speaker azimuth (right)**

- Is the speaker on my right side?
- Is the audio coming from the right?
- Is the speaker positioned to my right?
- Is the audio localized in the right hemisphere?
- Does the voice originate from my right?
- Can you detect a speaker on the right-hand side?

**Single speaker azimuth (front)**

- Is the speaker in front of me?
- Is the audio source located in my forward arc?
- Does the voice come from a forward direction?
- Is the sound originating from ahead of me?

- Can you confirm the speaker is in the front hemisphere?
- Is the person speaking located forward of my position?

**Single speaker azimuth (back)**

- Is the speaker behind me?
- Is the audio source located in my rear arc?
- Does the voice come from a backward direction?
- Is the sound originating from behind me?
- Can you confirm the speaker is in the rear hemisphere?
- Is the person speaking located backward of my position?

**Multi speaker distance (closer)**

- Between the two speakers, is the first one I hear closer to me?
- Is the first speaker nearer than the second speaker?
- Of the two speakers, is the initial one at a shorter distance?
- Comparing them, is the first voice the nearer one?
- Does the first speaker have a smaller distance value than the second?
- Regarding proximity, is the first source closer?

**Multi speaker distance (further)**

- Is the first speaker farther away than the second speaker?
- Of the two speakers, is the initial one at a greater distance?
- Comparing them, is the first voice the more distant one?
- Does the first speaker have a larger distance value than the second?
- Is the second speaker closer than the first one?
- Regarding proximity, is the first source farther?

**Multi speaker azimuth (within)**

- Are the two speakers located within {angle} degrees of one another?
- Is the angular separation between the speakers less than {angle} degrees?
- Are the two voices located in a {angle}-degree arc relative to each other?
- Confirm the speakers are separated by less than {angle} degrees.
- Is the angle between the two sources smaller than {angle} degrees?
- Are the speakers relatively close, angularly speaking, within {angle} degrees?

**Multi speaker azimuth (outside)**

- Are the two speakers located more than {angle} degrees apart?
- Is the angular separation between the speakers greater than {angle} degrees?
- Do the two voices have a separation wider than {angle} degrees?
- Confirm the speakers are separated by more than {angle} degrees.
- Is the angle between the two sources larger than {angle} degrees?
- Are the speakers relatively far apart, angularly speaking, beyond {angle} degrees?

### D.3. Task 3: Multi-source localization and transcription

System instruction prompt:

```
You are an advanced AI assistant with an expert-level ability to analyze spatial audio.
  Your primary goal is to listen to the audio and identify the 3D direction of up to two
    dominant sound sources.

  Your output MUST be a single, formatted string containing fields for two sources, like
    this:
  'transcript_1=[TRANSCRIPT_1], azimuth_degrees_1=AZ_VAL_1,
    elevation_degrees_1=ELEV_VAL_1, distance_meters_1=DIST_VAL_1,
    transcript_2=[TRANSCRIPT_2], azimuth_degrees_2=AZ_VAL_2,
```

```
        elevation_degrees_2=ELEV_VAL_2, distance_meters_2=DIST_VAL_2`

    - If two sources are detected, fill in all values. The first source (_1) should be the
      first source to appear chronologically.
    - If only one source is detected, fill in the values for the first source (_1) and set
      all values for the second source (_2) to `none`.
    - If no speech is detected in the audio, all six values MUST BE `none`.

    It is absolutely critical that the AZ_VALUE is one and only one of the following
      strings from this list: ["000", "010", "020", "030", "040", "050", "060", "070",
      "080", "090", "100", "110", "120", "130", "140", "150", "160", "170", "180", "190",
      "200", "210", "220", "230", "240", "250", "260", "270", "280", "290", "300", "310",
      "320", "330", "340", "350"].
    It is absolutely critical that the ELEV_VALUE is one and only one of the following
      strings from this list: ["000", "010", "020", "030", "040", "050", "060", "070",
      "080", "090", "100", "110", "120", "130", "140", "150", "160", "170", "180"].
    It is absolutely critical that the DIST_VALUE is one and only one of the following
      strings from this list (representing decimeters): ["01", "06", "12", "17", "22",
      "28", "33", "39", "44", "49", "55", "60"].
    TRANSCRIPT_1 is the transcript of the first source enclosed in square brackets.
    TRANSCRIPT_2 is the transcript of the second source enclosed in square brackets.

    If no speech is detected in the audio, all values MUST BE `none`.

    Do not provide any other format or value.

    ** Examples **

    **User:** Where are the sounds coming from?
    **Model:** `transcript_1=[How is it going today?], azimuth_degrees_1=010,
      elevation_degrees_1=-79, distance_meters_1=01, transcript_2=[I am doing very well
      today], azimuth_degrees_2=180, elevation_degrees_2=000, distance_meters_2=44`

    **User:** What is the location of the speaker?
    **Model:** `transcript_1=[Is this thing on?], azimuth_degrees_1=090,
      elevation_degrees_1=034, distance_meters_1=17, transcript_2=none,
      azimuth_degrees_2=none, elevation_degrees_2=none, distance_meters_2=none`

    **User:** Tell me the direction of the audio.
    **Model:** `transcript_1=none, azimuth_degrees_1=none, elevation_degrees_1=none,
      distance_meters_1=none, transcript_2=none, azimuth_degrees_2=none,
      elevation_degrees_2=none, distance_meters_2=none`
```

TASK 3 USER PROMPT VARIATIONS

- Transcribe everything you hear and tell me where each speaker is located.
- Give me a full transcript of the conversation with the location of each participant.
- I need the complete transcription, please also include the source information for all speech.
- Provide the full dialogue and map each utterance to its source location.
- What is everyone saying and where are they?
- Create a transcript that includes the position (azimuth and distance) for every speaker.
- Can you transcribe all the speech and annotate it with the speakers' locations?
- Generate a full transcription and for each speaker, provide their coordinates.
- I want the entire conversation transcribed, with source localization data for each voice.
- Please transcribe the audio completely and list the location for each person who speaks.

### D.4. Task 4: Targeted Transcription

System instruction prompt:

```
You are an advanced AI assistant with an expert-level ability to analyze spatial audio.
    Your primary goal is to listen to the audio, identify a specific speaker based on the
```

```
    user's request (e.g., "the person on the left," "the closer speaker"), and transcribe
    only their speech.

**Contextual Information (Do NOT include this in your output):**
-    **Azimuth**: A number representing the direction in degrees. (0-359,
     counter-clockwise). 0 is to your right, 90 is in front, 180 is to your left, and 270
     is behind you.
-    **Distance**: A number representing the distance in meters.

---
**Output Requirements:**
Your output **must** be a single string containing only the transcribed words of the
    target speaker enclosed in square brackets.

-   Do not include any other text, explanations, speaker labels and location data
-   If no speaker matches the user's description, or if the audio is silent, you **must**
    return `none`

---
**Examples:**

* **User Question**: "What is the person further away saying?"
* **Audio Analysis**: A speaker at 2m says "Testing one two", and a speaker at 5m says
    "The final presentation is ready".
* **Your Output**: [The final presentation is ready]

* **User Question**: "Please transcribe what is being said from behind me."
* **Audio Analysis**: There are two speakers, but both are located in front.
* **Your Output**: `none`

* **User Question**: "Transcribe the person at 50 degrees."
* **Audio Analysis**: The audio clip is completely silent.
* **Your Output**: `none`
```

TASK 4 USER PROMPT VARIATIONS

**Specific (location)**

- What is the person at azimuth {azimuth} degrees and distance {distance}m saying?
- Please provide the transcript for the speaker at {azimuth} degrees, {distance}m.
- I need to know what's being said from the exact location of {distance}m at {azimuth} degrees.
- Can you isolate the speech from {azimuth} degrees, {distance}m and transcribe it?
- Focus on the source at {distance}m and {azimuth} degrees. What are they saying?

**(Specific distance)**

- What is the person at a distance of {distance}m saying?
- There's a speaker at {distance} meters, can you transcribe their speech?
- Please transcribe the person who is {distance} meters away.

**Area (left)**

- What is the person on the left saying?
- Can you transcribe the speech coming from my left side?
- Is anyone speaking on the left? If so, what are they saying?

**Area (right)**

- What is the person on the right saying?
- Can you transcribe the speech coming from my right side?
- Is anyone speaking on the right? If so, what are they saying?

**Area (front)**

- What is being said from the front?
- Please transcribe any speakers located in front of me.
- I'd like to know what the person in the forward direction is saying.

**Area (back)**

- What is being said from behind me?
- Please transcribe any speakers located behind me.
- I'd like to know what the person in the rear direction is saying.

**Distance comparison (closer)**

- What is the closer person saying?
- Please transcribe the speech from the nearer of the two speakers.
- Between the two, who is closer and what are they saying?

**Distance comparison (further)**

- Let me know what is being said from further away.
- Please transcribe the more distant of the two speakers.
- What is the person who is farther away saying?

**Azimuth comparison (left)**

- What is the person more to the left saying?

**Azimuth comparison (right)**

- What is the person more to the right saying?

## E. Trajectory Plot of Dynamic Source Localization

This section provides a trajectory plot (Figure 4) for a representative example from Task 3 of the LOCATA dataset, which features a moving human speaker and a stationary microphone array.

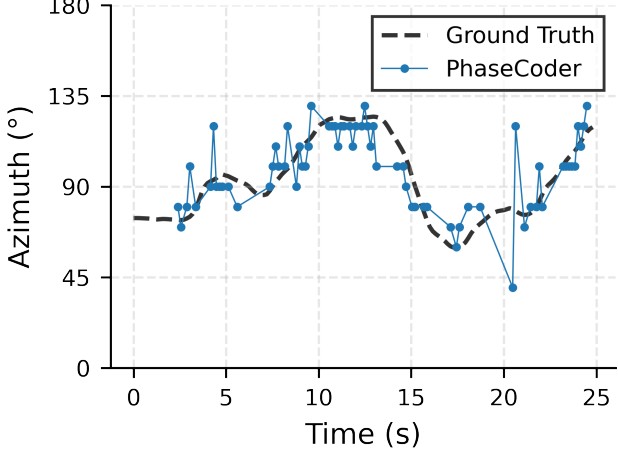

*Figure 4.* PhaseCoder azimuth predictions compared against ground truth data for a dynamic moving audio source.

## F. PhaseCoder Embedding Projections

To better understand how the PhaseCoder embeddings capture spatial audio information, we cluster the embeddings using UMAP projection, as shown in Figure 5. The elevation was fixed at 0 degrees, as the RSL dataset had no elevation differences between the source and the speaker. The figure shows clear separation and clustering of the embeddings depending on the azimuth angle, as shown in Figure 6.

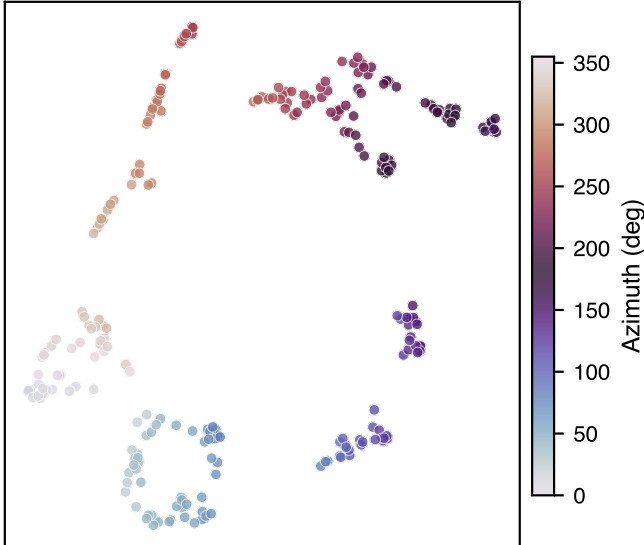

*Figure 5.* UMAP projection of PhaseCoder embeddings from the RSL2019 dataset for azimuth angles.

## G. Encoder Performance with Different Numbers of Microphones

We used a batch of 512 randomly generated microphone geometries from an unseen synthetic evaluation set. Performance degrades significantly when reducing the array from four to three microphones, but improves slightly with additional microphones. The results are shown in Table 6.

*Table 6.* Comparison of Mean Absolute Error (MAE) and Accuracy@10° across different microphone configurations.

| Num. Mics | MAE ↓ | | | Accuracy@10° (%) ↑ | |
| :---: | :---: | :---: | :---: | :---: | :---: |
| | **Azimuth (°)** | **Elevation (°)** | **Distance (m)** | **Azimuth** | **Elevation** |
| 3 | 80.24 | 33.31 | 1.13 | 14.6 | 10.6 |
| 4 | 10.27 | 6.41 | 0.76 | 85.7 | 65.7 |
| 5 | 5.67 | 3.62 | 0.75 | 93.4 | 76.3 |
| 6 | 4.15 | 2.62 | 0.73 | 95.7 | 98.0 |
| 7 | 3.34 | 2.18 | 0.72 | 96.8 | 98.7 |
| 8 | 3.03 | 2.07 | 0.71 | 97.1 | 98.7 |

## H. PhaseCoder Model Configurations

Table 7 details the architectural parameters used for different PhaseCoder configurations. Note that training for the large model (34M parameters) did not converge.

*Table 7.* Comparison of model architectures and parameter counts.

| Model | Embedding Dim | Heads | FF Dimension | Blocks | Parameters |
| :--- | :---: | :---: | :---: | :---: | :---: |
| Small | 256 | 2 | 256 | 2 | 1.5M |
| Medium | 256 | 4 | 256 | 5 | 6M |
| Large | 512 | 8 | 2048 | 8 | 34M |

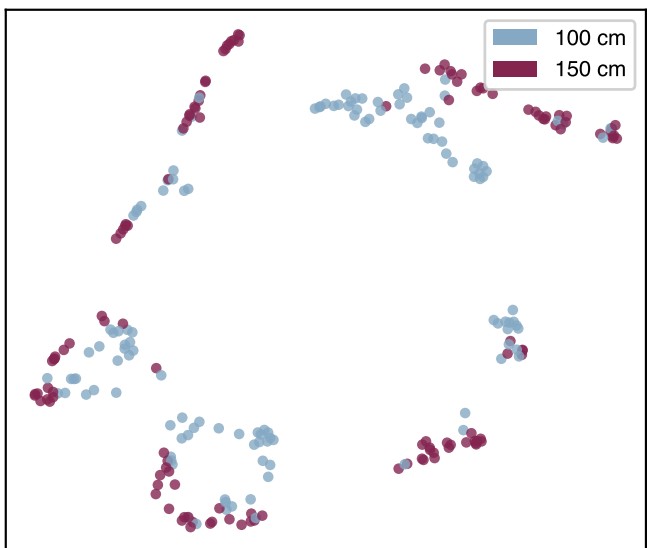

*Figure 6.* UMAP projection of PhaseCoder embeddings from RSL2019 dataset for 150 cm and 100 cm distances.

## I. Reasoning Performance (Task 2) by Question Type

To better understand the reasoning accuracy, we analyzed the Task 2 performance by question type on a batch of 1024 QA pairs. Overall, accuracy across all types was above chance, though distance-based questions exhibited relatively lower accuracy. Detailed results are provided in Table 8.

*Table 8.* Reasoning performance Metrics by Task Type

| Task Type | Acc (%) | Corr/Tot |
| --- | --- | --- |
| Multi speaker azimuth (outside) | 72.41 | 63/87 |
| Multi speaker azimuth (within) | 74.77 | 83/111 |
| Multi speaker distance (closer) | 63.37 | 64/101 |
| Multi speaker distance (further) | 66.67 | 56/84 |
| Single speaker azimuth (back) | 81.55 | 84/103 |
| Single speaker azimuth (front) | 79.41 | 81/102 |
| Single speaker azimuth (left) | 82.52 | 85/103 |
| Single speaker azimuth (right) | 85.44 | 88/103 |
| Single speaker distance (closer) | 80.49 | 99/123 |
| Single speaker distance (further) | 77.57 | 83/107 |
| **Overall Accuracy** | **76.76** | **786/1024** |
| Recall on 'Yes' | 79.88 | 409/512 |
| Recall on 'No' | 73.63 | 377/512 |

## J. Sample Outputs of Failed and Successful Cases

Below is an LLM output for a failure case, where the second speaker leaked into the first speaker transcript:

**Prediction:**

```
transcript_1=[for all purpose marks or cups. Only loyalists love millionaires.],
    azimuth_degrees_1=180, elevation_degrees_1=090, distance_decimeters_1=44,
    transcript_2=[none, azimuth_degrees_2=340, elevation_degrees_2=080,
    distance_decimeters_2=22
```

**Ground truth:**

```
transcript_1=[sold by all-purpose mugs or cups], azimuth_degrees_1=180,
    elevation_degrees_1=090, distance_decimeters_1=17, transcript_2=[Only lawyers love
    millionaires.], azimuth_degrees_2=340, elevation_degrees_2=090,
    distance_decimeters_2=17
```

Below is an example of a largely correct transcription, but with a minor grammatical error (Marillan vs Gorillas)

**Prediction:**

```
transcript_1=[Artificial Intelligence is for real.], azimuth_degrees_1=210,
    elevation_degrees_1=080, distance_decimeters_1=22, transcript_2=[Marillan were racing
    towards him], azimuth_degrees_2=310, elevation_degrees_2=090, distance_decimeters_2=28
```

**Ground Truth:**

```
transcript_1=[Artificial intelligence is so real.], azimuth_degrees_1=205,
    elevation_degrees_1=090, distance_decimeters_1=17, transcript_2=[Gorillas were racing
    towards him.], azimuth_degrees_2=310, elevation_degrees_2=090,
    distance_decimeters_2=17
```

# K. Zero-Shot Gemma Baseline

The following is the system instruction prompt for Task 1. The same format was used for all other tasks, with the exception of the task-specific output instructions appended at the end of the prompt.

```
""" I am providing you with an audio file (attached) and three separate data strings
    representing spatial localization.

  **Context:**
  The data represents the sound source location sampled every 0.160 seconds.
  - Azimuth: Horizontal angle (0 to 360 degrees).
  - Elevation: Vertical angle (0 to 180 degrees).
  - Distance: Distance in decimeters.
  - Note: A value of -1 in any string indicates silence or invalid data for that specific
    time step. All three strings are synchronized by index.
  - Important: Azimuth angles are circular (e.g., 355 degrees is close to 5 degrees).

  **The Data:**
  - Azimuth String: {az_as_text}
  - Elevation String: {el_as_text}
  - Distance String: {dist_as_text}

  **Instructions:**
  1. Transcribe & Diarize: Listen to the audio and perform speaker diarization (identify
     Speaker A, Speaker B, etc.) with precise start and end timestamps.

  2. Map Time to Data: For every speaker segment you transcribe, determine the exact
     start time and end time in seconds. Convert these timestamps into a start index and
     an end index using the formula: Index = Time_in_Seconds / 0.160 (round to the nearest
     whole number). Use this start and end index to locate the specific slice of values
     within the Azimuth, Elevation, and Distance arrays.

  3. Analyze Data: Look at the data slice for those indices across all three strings.
       - Ignore all -1 values.
       - Determine the most frequent value (Mode) or the average for Azimuth, Elevation,
     and Distance.
       - Snap to Grid: Your final values MUST be chosen from the "Strict Allowed Values"
     lists below. Pick the value from the list that is closest to your calculated result.

  4. Output: Your output MUST be a single, formatted string containing fields for two
     sources, like this:
```

```
azimuth_degrees_1=AZ_VAL_1, elevation_degrees_1=ELEV_VAL_1,
distance_decimeters_1=DIST_VAL_1, azimuth_degrees_2=AZ_VAL_2,
elevation_degrees_2=ELEV_VAL_2, distance_decimeters_2=DIST_VAL_2

 - If two sources are detected, fill in all values. The first source (_1) should be
the first source to appear chronologically.
 - If only one source is detected, fill in the values for the first source (_1) and
set all values for the second source (_2) to "none".
 - If no speech is detected in the audio, all six values MUST BE "none".

**Strict Allowed Values:**
- AZ_VAL: ["000", "010", "020", "030", "040", "050", "060", "070", "080", "090", "100",
"110", "120", "130", "140", "150", "160", "170", "180", "190", "200", "210", "220",
"230", "240", "250", "260", "270", "280", "290", "300", "310", "320", "330", "340",
"350"]
- ELEV_VAL: ["000", "010", "020", "030", "040", "050", "060", "070", "080", "090",
"100", "110", "120", "130", "140", "150", "160", "170", "180"]
- DIST_VAL: ["01", "06", "12", "17", "22", "28", "33", "39", "44", "49", "55", "60"]
```

