# OpenReview forum: "PhaseCoder: Microphone Geometry-Agnostic Spatial Audio Understanding for Multimodal LLMs"
_ICML.cc/2026/Conference — ICML 2026 regular_

### Official Review · Reviewer_8W7x · 2026-03-05

**Soundness:** 3
**Presentation:** 3
**Significance:** 3
**Originality:** 3
**Overall Recommendation:** 5
**Confidence:** 5

**Summary:**

This paper proposes PhaseCoder, a microphone-array agnostic spatial audio encoder that takes raw multichannel audio and microphone coordinates and produces spatial audio tokens that can be injected into a multimodal LLM for spatial reasoning and spatially cued/targeted transcription. The work strives to explore the concept of universal spatial audio representations that can transfer across arbitrary mic arrays, addressing a key limitation of many prior spatial audio llm pipelines that are tied to fixed geometries. This study intends to outline the domain of geometry-invariant spatial audio understanding for embodied agents and multimodal assistants.

**Compliance With Llm Reviewing Policy:**

Affirmed.

**Ethical Review Concerns:**

No such concern

**Final Justification:**

My concern is addressed and I would encourage the author to incorporate the rebuttal points into the revised version. I have raised my score to 5.

**Key Questions For Authors:**

1. One biggest question is the large gap between the simulated dataset and the real dataset. I originally thought that there would be large gap in the localization since microphone array/geometries would actually introduce lots of channel diversity to influence the localization accuracy. However, it seems that the localization accuracy just drop by several degress which is acceptable. However, what surprised me is the large gap in WER between the simulated dataset and the real dataset. Can you explain why? I think the WER should be too bad between the simulated one and the real one since they rely more on signal semantic features rather than signal level features, which is critical for localization.

**Limitations:**

See weaknesses and questions.

**Strengths And Weaknesses:**

Strengths:
1. Clear problem and strong practical motivations. Most devices have mic arrays but model pipelines are typically geometry-locked; the paper directly targets this limitation with a geometry-agnostic encoder and coordinate-conditioned positional encoding
2. Solid encoder results on real-world localization benchmarks: The PhaseCoder-6M model is competitive and even improves over GI-DOAEnet on LOCATA metrics, while remaining strong on RSL2019.
3. The network design to process the spatial audio is interesting. It is novel to blend the multi-channel microphone recordings and fuse with temporal informations.

Weaknesses:
1. There is a large gap between the simulated dataset and the real-dataset. In the table 2, the task 4 of WER for simuated dataset seems too good to be true (0%) in the simulated dataset, but it seems too bad in the real dataset (42.86%). Even the Gemma basleine is 35.41%, I am just wondering whether in reality this recognition quality is good enough to use.
2. Multi-speaker capability remains limited vs the direction of the field: The pipeline largely handles up to two speakers without overlap and the paper itself frames “single dominant speech source” as the main current scope. Given that prior systems already explore multi-speaker transcription/diarization with spatial cues on wearable/array platforms (SING: Spatial Context in Large Language Model for Next-Gen Wearables ICML2025), it would strengthen the contribution to more directly benchmark against multi-speaker ASR/diarization baselines or at least run overlapping-speech variants.
3. I feel that the evlauation on baseline is limited. The paper did not compare with BAT. Besides, the paper did not discuss about the cascated signal processing baseline like first doing microphone array beamforming + ASR or beamforming + diarization + ASR. And it would also be interesting to first do beamforming based on the array geometreis and send the resulting probing signal into a transformer.
4. Broaden related work and positioning. In additiona to spatial audio LLM reasoning, the paper would benefit from discussing the emerging trend of spatial audio generation/editing pipelines and how geometry-agnostic spatial understanding tokens could serve as control signals or evaluation tools for those systems.

Sun, Peiwen, et al. "Both ears wide open: Towards language-driven spatial audio generation." arXiv preprint arXiv:2410.10676 (2024).

Lan, Zitong, Yiduo Hao, and Mingmin Zhao. "Guiding audio editing with audio language model." arXiv preprint arXiv:2509.21625 (2025).

Kim, Jaeyeon, Heeseung Yun, and Gunhee Kim. "Visage: Video-to-spatial audio generation." arXiv preprint arXiv:2506.12199 (2025).

---

> ### Author Rebuttal · Authors · 2026-03-30
>
> Thank you for the insightful review and recognizing the contributions of this work.
>
> > One biggest question is the large gap between the simulated dataset and the real dataset.
>
> Fundamentally, the RSL2019 dataset is much more challenging than LibriSpeech that was used for the synthetic dataset. We listened to a lot of raw samples from RSL2019 and many are difficult to comprehend for the human ear. Confirming that, the state-of-the-art multimodal Gemini 3 model got 27 % WER. Also, it is possible that LLMs has seen LibriSpeech in training, as it is the most common ASR dataset. It is unlikely they encountered RSL2019 before.
> Regarding the 0 % WER, it is a **median** WER. The mean WER was 5 - 10 %, meaning that the LLMs still made many mistakes, but most often transcribed perfectly.
>
> > Multi-speaker capability remains limited vs the direction of the field:
>
> Since the submission, we retrained the PhaseCoder model to work with overlapping speakers by modifying the training to localize two overlapping speakers.  The model was trained in three stages: 1 speaker →2 speaker funetuning → noise finetuning. Evaluating on the synthetic dataset shows a decrease in performance from 86 % to 72% in azimuth MAE.  We will add preliminary results with this model to the Appendix.
>
> > I feel that the evaluation on baseline is limited. The paper did not compare with BAT
>
> We have added a 2-stage cascaded baseline (Standalone PhaseCoder coordinates fed as text prompts into Gemma). The baseline underperforms the fine-tuned model. See Reviewers zSY6 for more details.
>
> Unfortunately, we could not use BAT as a baseline, as BAT worked with two binaural microphones and our model used 3 or more. However, we have included a computational benchmarking against BAT in our response to Reviewer Gtk5) to show that PhaseCoder is more memory and compute efficient.
>
> We considered a baseline to beamform the audio first and feed it into the Gemma mono encoder. However, it is still an open research question on how to make a controllable microphone agnostic beamformer, and such a baseline would be more reflective of beamformer performance.
>
> > Broaden related work and positioning.
>
> We will add a related work section on editing and generation to broaden the scope. We are currently experimenting with the spatial audio generation, so that would be a very relevant addition.
>
> Recent spatial audio generation and editing systems further motivate the need for rich spatial representations. SpatialSonic (Sun et al., ICLR 2025) conditions a latent diffusion model on spatial-aware encoders and azimuth state matrices, ViSAGe (Kim et al., ICLR 2025) autoregressively generates first-order ambisonics from silent video using directional and visual guidance, SALM (Hu et al., 2025) learns structured audio embeddings to support both spatial audio understanding and flexible spatial audio editing, and SmartDJ (Lan et al., 2025) uses an audio language model to decompose high-level instructions into atomic spatial edit operations which are executed by a diffusion model. These works suggest that universal spatial audio encoders could serve not only reasoning but also as control signals for generation and editing pipelines.
>
> References:
>
> **SpatialSonic**: Sun, P., Cheng, S., Li, X., Ye, Z., Liu, H., Zhang, H., Xue, W., & Guo, Y. (2025). Both ears wide open: Towards language-driven spatial audio generation. In Proceedings of the International Conference on Learning Representations (ICLR 2025).
>
> **ViSAGe**: Kim, J., Yun, H., & Kim, G. (2025). ViSAGe: Video-to-spatial audio generation. In Proceedings of the International Conference on Learning Representations (ICLR 2025).
>
> **SALM**: Hu, J., Cao, Y., Wu, M., Luo, Z., & Yang, J. (2025). SALM: Spatial audio language model with structured embeddings for understanding and editing. arXiv preprint arXiv:2507.16724.
>
> **SmartDJ**: Lan, Z., Hao, Y., & Zhao, M. (2025). Guiding audio editing with audio language model. arXiv preprint arXiv:2509.21625.

---

> > ### Author Rebuttal · Reviewer_8W7x · 2026-04-03
> >
> > Thank you for the detailed response. I suggest the authors incorporate the discussion of experiemtn and related work into the revised version. good luck!

---

### Official Review · Reviewer_Gtk5 · 2026-03-08

**Soundness:** 3
**Presentation:** 2
**Significance:** 2
**Originality:** 3
**Overall Recommendation:** 3
**Confidence:** 5

**Summary:**

This paper addresses the gap that multimodal LLMs lack spatial audio understanding while existing spatial audio models are restricted to fixed microphone geometries, proposing PhaseCoder—a transformer-only, microphone geometry-agnostic spatial audio encoder. Taking raw multichannel audio and microphone coordinates as inputs, PhaseCoder extracts phase-based spatial features, fuses three positional embeddings, and uses a lightweight 5-block Transformer to generate spatial audio tokens. Integrated with Gemma 3n via a 2-layer MLP projection and LoRA fine-tuning (r=8, α=16), spatial tokens are prepended to audio tokens with BSA/ESA markers. The fine-tuned model is validated on four synthetic tasks (localization, spatial reasoning, spatial/targeted transcription) and real-world datasets, showing significant reasoning gains (76.76% accuracy on yes/no questions) and effective targeted transcription, though distance estimation underperforms due to underdetermined acoustic parameters.
The work strives to explore the concept of universal spatial audio encoding for LLMs, breaking hardware geometry constraints. This study intends to outline the domain of spatial audio understanding for multimodal LLMs, with societal impacts including enhanced accessibility and embodied intelligence, while identifying limitations like static source assumptions and single-speaker focus for future improvement.

**Compliance With Llm Reviewing Policy:**

Affirmed.

**Final Justification:**

This article is a well-conducted engineering study. However, from the theoretical framework and innovative concepts introduced at the project's inception, to the interpretation of experimental results in light of that framework, and to the rigorous formulation of evaluation methods, ablation studies, and other aspects, the work remains insufficiently refined. While this paper qualifies as an excellent technical report, it still exhibits significant theoretical gaps when evaluated as an academic publication.

Especially for reviewers outside the spatial audio field, eliminating the influence of microphone arrangement appears to be a very attractive innovation. However, upon closer inspection, such an innovation is likely completely nonexistent. Therefore, I maintain my opinion regarding the edge rejection.

**Key Questions For Authors:**

1. PhaseCoder’s performance degrades drastically with <4 microphones (3 mics: Azimuth Acc@10=14.6%), yet consumer devices often have 2–3 mics—**what architectural or training modifications are you investigating to improve generalization to sparse microphone arrays (2–3 mics), and what is the minimum number of mics you believe PhaseCoder can effectively support with these changes?
2. The model assumes static sound sources and free-floating omnidirectional microphones, ignoring device-specific acoustic effects (baffling, diffraction) and microphone beampatterns—have you conducted preliminary experiments with simulated device-specific acoustic properties (e.g., COMSOL-generated transfer functions) or moving sources, and how much does the current model’s performance degrade in these more realistic scenarios?
3. The choice of input representation for spatial audio requires prudent consideration, comparison, experimentation, and justification.  It is highly desirable for the authors to provide additional experiments and discussions to elaborate on the rationale behind their choice of input format. Is it possible for these diverse formats to share a universal representation or feature space?
4. Granularity of Supervision: Why was a coarse-grained discrete classification chosen as the supervisory objective, rather than a target that more closely aligns with continuous spatial geometry? Does this design choice effectively impose a "ceiling" on the embedding's representational capacity?
5. Temporal Alignment: How natural is the temporal alignment of the spatial tokens? Specifically, is there any information "folding" or misalignment between the 250ms encoding window and the 160ms cadence of the LLM’s audio tokens?
6. Interface Design: What is the rationale behind prepending all spatial tokens before the audio tokens? Have the authors compared this with alternative integration methods, such as interleaving, cross-attention, or shallower fusion (e.g., direct addition)?
7. Architecture vs. Training Recipe: A primary concern is whether the success of the model stems from the architecture itself or is heavily dependent on a highly specific "curriculum learning recipe." The training process described is remarkably complex: the spatial encoder requires 670k steps on clean speech followed by 30k steps of noisy fine-tuning, while the LLM fine-tuning utilizes a 5-stage curriculum. The authors explicitly state that "without the curriculum, the model struggled to learn." If a method only stabilizes under such meticulous training choreography, it becomes difficult to discern whether the contribution lies in the architecture or the specific training heuristics.
8. Complex Scenarios (Overlap): Regarding more challenging scenarios, such as overlapping sound sources, I would encourage the authors to provide some preliminary experimental results. Even if the performance is suboptimal, these insights would be highly valuable for the community's exploration of this frontier.
9. Pure Tone and Chirp Sweep Localization: Evaluate the encoder's MAE using pure tones across different frequencies and a logarithmic sine sweep (100 Hz to 8 kHz). Does the localization accuracy collapse for frequencies above the spatial aliasing
limit (> 1 kHz)?
10. Band-limited Speech Evaluation: Evaluate the localization performance on the synthetic or RSL2019 dataset using low-pass filtered speech (< 1 kHz) versus highpass filtered speech (> 2 kHz). A significant performance drop in the high-pass condition would reveal the model's inability to utilize high-frequency spatial cues due to the lack of modeled acoustic shadowing.
11. DRR-Controlled Distance Estimation: Audio-only distance estimation is notoriously ill-posed. To prove the model learned spatial acoustics rather than just amplitude scaling (loudness), test the distance estimation MAE on sources with normalized RMS energy but varying Direct-to-Reverberant Ratios (DRR).

**Limitations:**

yes

**Strengths And Weaknesses:**

## Soundness
**Strengths**: The work is technically rigorous, with PhaseCoder’s design rooted in valid signal processing and DL principles, and LLM integration following SOTA multimodal practices. Experiments are well-designed, comparing PhaseCoder to GI-DOAEnet on RSL2019/LOCATA with standard metrics, and four progressive synthetic tasks for LLM validation.
**Weaknesses**: Lacks real-world consumer device data validation and ablation studies for key components (positional embeddings, LoRA hyperparameters). No quantitative analysis of device-specific acoustic effects on performance.
## Presentation
**Strengths**: Logically structured with a coherent narrative, thorough prior work contextualization, and detailed experimental setup/docs. Core contributions are clearly articulated upfront, with an impact statement linking technical work to societal value.
**Weaknesses**: Minor reproducibility gaps (underdocumented implementation details, no code release), critical task details buried in appendices, limited spatial token behavior visualization, and small notation inconsistencies.
## Significance
**Strengths**: Addresses a critical multimodal LLM gap (audio-blind to spatial info) and advances both spatial audio processing and LLM research. Delivers broad practical/societal impact (assistive tech, consumer AI, embodied intelligence) and unlocks new research directions for multi-sensory fusion and dynamic spatial audio understanding.
**Weaknesses**: Focuses solely on speech sources, no scalability evaluation on larger LLMs, and no novel theoretical ML contributions.
## Originality
**Strengths**: High originality via creative combination of microphone-agnostic localization, Transformer encoding and LoRA fine-tuning, with well-articulated reasoning. Removes fixed microphone geometry assumptions from prior work, introduces four novel LLM spatial audio tasks, and modifies positional embeddings for better spatial encoding. Clear distinction from SOTA work is provided.
**Weaknesses**: No entirely new architectural components, unoriginal distance estimation methods, and standard synthetic data generation techniques.

---

> ### Author Rebuttal · Authors · 2026-03-30
>
> Thank you for detailed feedback on the paper.
>
> > Weaknesses: Minor reproducibility gaps
>
> We will attempt to  open source the model, given that we can obtain permission from our institution.
>
> > Weaknesses: Lacks real-world consumer device data validation
>
> The evaluations were done on real recorded device data: microphone array in RSL2019 and robot head in LOCATA.
>
> > What architectural or training modifications are you investigating to improve generalization to sparse microphone arrays?
>
> We are investigating how to encode and embed device or head related transfer functions (HRTFs) to allow the model to function with sparse microphone arrays. This is future work.
>
> > Moving sources experiments
>
> We conducted additional evaluations for dynamic source evaluation. More details in rebuttal for Reviewer NeYq. We used Task 3 in the LOCATA dataset that has a static microphone array and a moving human speaker. The results look promising and show that the model can localize moving sources with about 10 degree error. The azimuth angle over time plots also show consistent tracking of moving sources.
>
> > The choice of input representation for spatial audio
>
> Initially we experimented with different input features for the encoder and found the magnitude/phase representation to converge the fastest during training. We trained three separate models to 1.5M steps (25 days) with:  (1) **real/imaginary** , (2) **magnitude only** and (3) **IPD** (interaural phase differences) features and the model was not able to converge
>
> > Granularity of Supervision
>
> Our reasons for choosing categorical (cross-entropy loss) vs regression for localization:
> * The majority of  state-of-the-art approaches, especially related to spatial audio LLM interfacing (e.g., Spatial-AST) use categorical loss. Furthermore, our training recipe and backbone architecture follow the ViT and AST (Audio Spectrogram Transformer) paradigms, which use cross-entropy objectives.
> * We can directly compare PhaseCoder  to microphone-agnostic  GI-DOAENet, which uses categorical loss.
> * Real-world sound localization is probabilistic, making a statistical distribution of locations more fitting than an exact regression target.
>
> Regarding the “ceiling” on representational capability, current LLMs operate in discrete token space so even if encoder operates in continuous space, the outputs will need to discretized
>
> > Temporal Alignment
>
> Temporal alignment issues would result in transcript swapping between two speakers. We did not observe that in the fine-tuned model.
>
> > Interface Design
>
> We did not interleave the spatial tokens with audio to follow Gemma input structure. In Gemma, the audio tokens are placed between beginning and end of audio control tokens, so interleaving with our tokens damages existing audio capabilities. Our initial experiments confirmed that. Also, we avoided methods like cross-attention with audio encoder and/or Gemma as those would be significant changes to existing LLM.
>
> > Architecture vs. Training Recipe
>
> In the audio language models, multistage training with LoRA is standard for alignment of spatial audio with LLMs. All the state-of-the art papers on the topic follow a similar procedure. (e.g, BAT). We experimented with simpler training, but did not find those effective. Also, we want to highlight that the core contribution of the paper is microphone embeddings, and not the LLM alignment. The alignment was done to demonstrate that LLM can use our embeddings. Effective multimodal training and alignment of LLMs is still an open research question.
>
> > Complex Scenarios (Overlap)
>
> Since the submission, we retrained the PhaseCoder model to work with overlapping speakers by modifying the training to localize two overlapping speakers.  The model was trained in three stages: 1 speaker →2 speaker funetuning → noise finetuning. Evaluating on the synthetic dataset shows a decrease in performance from 86 % to 72% in azimuth MAE.  We will add preliminary results with this model to the Appendix.
>
> > Pure Tone and Chirp Sweep Localization:
>
> Unfortunately, pure tones and chirp experiments are not possible due to fundamental design limitations.   The model is specifically trained to localize speech, and ignore noise and non-speech. Pure tones and sweeps will be classified as noise and will be ignored by the model.
>
> > Band-limited Speech
>
> The current model would not work well with band-limited signals because it was only trained to see full 16 kHz speech. For the model to work with band-limited speech, the band-limited signal augmentations would have to be done in training. As this would be a significant undertaking, it is not possible to do in the rebuttal timeline, so we will do that for revised version.
>
> > DRR-Controlled Distance Estimation
>
> During training the gain was varied randomly between -5 to 15 dB, to avoid the model from relying solely on amplitude.  We tested the model with normalized amplitudes and did not see much divergence on distance estimation

---

> > ### Author Rebuttal · Reviewer_Gtk5 · 2026-04-03
> >
> > I appreciate the authors' response. However, one core issue remains that I believe requires resolution:
> > A central premise of this paper is that, when converting an arbitrary microphone array into an Ambisonics or other array-agnostic spatial representation, "device-specific beamformers are required." Yet, under the idealized sensing assumptions adopted in this paper, the notion of "device-specific" here is actually open to question. This is because, within this specific setting, the reason such front-ends are considered "device-specific" is primarily that they require knowledge of the array geometry—specifically, the microphone positions—in order to construct the corresponding encoding or beamforming operators. However, the authors' proposed method itself also explicitly utilizes microphone coordinates as an input. In other words—at least within the context of the idealized array model discussed in this paper—both categories of methods rely on the exact same array-specific geometric information; the only distinction lies in how that information is utilized: one approach employs it within an explicit spatial encoding front-end, while the other feeds it into a learned encoder. Therefore, could the authors please further clarify: what concrete, practical advantages does their method truly offer compared to these Ambisonics/beamforming front-ends that operate with known geometry? If the so-called "device-specific" nature described here is, in essence, merely "geometry-dependent," then the core advantage claimed by the paper on this basis would appear to lose much of its practical significance.
> > Consequently, the authors need to undertake careful consideration, comparison, experimentation, and explication regarding the choice of representation for the spatial audio input. This is a non-trivial issue, though conducting such a comparison at the PhaseCoder training level (without integrating an LLM) does not appear to be particularly difficult. If feasible, I hope the authors can supplement the paper with relevant experiments and explanations, elaborating in detail on the rationale behind their choice of input format. Furthermore (though this is not intended as a condition for the paper's acceptance): is it conceivable that various formats could potentially share a common representation or feature space? I would be very interested to hear the authors' team's thoughts on this particular question.

---

> > > ### Author Response · Authors · 2026-04-03
> > >
> > > We appreciate the reviewer’s continued engagement and insightful follow-up questions. You raise excellent points regarding Ambisonics, input representations, and format-agnostic spaces, and we welcome the opportunity to expand on those topics.
> > >
> > > > Could the authors please further clarify: what concrete, practical advantages does their method truly offer compared to these Ambisonics/beamforming front-ends that operate with known geometry?
> > >
> > > * **Efficiency.** Ambisonics will require two model pipeline, while our approach uses just one model end-to-end.  First, a classical or neural beamformer will be required to process the raw audio to make the 4 ambisonic audio channels. Second, a model that compresses the 4-channel output into embeddings. From a practical standpoint, one model that works on raw audio and produces embeddings directly is more attractive for compute, training, deployment and accuracy. Ambisonics is just an intermediate format here.
> > >
> > > * **Calibration** Beamforming from an arbitrary mic array to ambisonics requires us to obtain the array transfer functions of the device. This either requires COMSOL simulations or anechoic chamber measurements, both of which are expensive and time consuming, compared to obtaining just the microphone positions.
> > >
> > > * **Number of Mics** Beamforming to FOA needs at least 4 mics (https://arxiv.org/html/2402.17362v1). PhaseCoder does work with 3 mics as well.
> > >
> > > * **Accuracy.** We conducted additional experiments with the STARSS23 dataset (https://arxiv.org/abs/2306.09126), which has both Ambisonics and raw 4-mic array data for the same acoustic scenes. We compare PhaseCoder to state-of-the-art ambisonics approach (Sound Event Detection and Localization with Distance Estimation 2024, https://arxiv.org/pdf/2403.11827 ) which was specifically trained for evaluations on STARSS.
> > > Compared to the reported values in the SELD Ambisonics (2024) paper (Table 3), our approach achieves higher accuracy for azimuth angle, while not seeing STARSS during training. As this is a promising result, we will expand on this analysis in the revised paper.
> > >
> > > | Dataset | MAE Azimuth degrees (↓) | MAE Elevation degrees (↓)
> > > | :--- | :--- | :--- |
> > > | SELD Ambisonics (2024)  | 17.7 | - |
> > > | PhaseCoder  | **14.5** | 8.03 |
> > >
> > >
> > > > Input representations
> > >
> > > We agree that considering input representations are important. In the final revision of the paper, we will include our existing exploration results using (1) **magnitude/phase**, (2) **real/imaginary** , (3) **magnitude only** and (4) **IPD** representation to the paper.
> > >
> > > Below is a table of the result on synthetic evaluation set after training stage 1 of PhaseCoder. We used the same architecture, learning rate schedule and hyperparameters in all the experiments.
> > >
> > > We suspect that the degraded performance with other representations might be due to hyperparameters such as learning rate schedule, as transformer architecture is sensitive to that. In practice, we found it difficult to predict how the transformer will behave in training, and we can only sweep a limited amount of parameters, since the compute cost is high.  We show the results in the table below, which we could not provide earlier due to the response character limit.
> > >
> > > | Input Features | Azimuth accuracy % (↑) | MAE Azimuth (↓) | Elevation accuracy % (↑) | Distance accuracy % | Train steps |
> > > | :--- | :--- | :--- | :--- | :--- | :--- |
> > > | Mag / Phase  | **83.94** | **1.72** | **89.33** | **46.40** | 680 K |
> > > | Real / Imag  | 6.45 | 84.57 | 10.96 | 12.51 | 1.5 M |
> > > | Mag only | 7.47  | 84.58 | 16.47 | 28.56 | 1.5 M |
> > > | IPD | 11.56  | 84.40 | 18.75 | 29.19 |  1.5 M |
> > >
> > > >Formats agnostic representations
> > >
> > > Common representations have been on our mind in regards to future work. For example, we believe that the model could be trained to work as well with Ambisonic inputs. To do so we would need to add an “ambisonic” microphone embedding instead of geometry embedding and some ambisonic training examples. Furthermore, we could add various other formats such as binaural, 5.1 surround, etc, with the same process. Specific geometry embeddings could be used to indicate the formats of the inputs. Potentially such a model could even take mixed formats as input (e.g, raw mics + ambisonics).

---

### Official Review · Reviewer_NeYq · 2026-03-11

**Soundness:** 3
**Presentation:** 3
**Significance:** 3
**Originality:** 3
**Overall Recommendation:** 5
**Confidence:** 4

**Summary:**

This paper introduces a novel, transformer-only spatial audio encoder named PhaseCoder. The model takes raw multichannel audio and microphone coordinates as input to generate robust spatial embeddings (spatial audio tokens) that are agnostic to microphone geometry. By injecting these spatial tokens into the Gemma 3n multimodal large language model (LLM), the research team enables an LLM to perform complex spatial reasoning and targeted transcription tasks from an arbitrary microphone array. This approach surpasses the performance of existing microphone-invariant localization models on real-world datasets.

**Compliance With Llm Reviewing Policy:**

Affirmed.

**Final Justification:**

As stated in the Acknowledgement, the authors have addressed my concerns. I would maintain my positive score for this paper.

**Key Questions For Authors:**

#### 1. **Performance limits with very few microphones**: The model was trained with 3 to 8 microphones.  When the number of microphones is reduced to 2 (e.g., common dual-microphone arrays, TWS earbuds), can the model still retain basic directional capability or spatial reasoning ability?

#### 2. **Practicality of microphone coordinates**: In practical applications, how are the microphone coordinates obtained?  Must they be completely hardcoded/given?

#### 3. **Adaptive solutions for coordinate-less scenarios**: If microphone coordinates must be provided, how should this model be used in consumer scenarios where the exact microphone coordinates of the user's device cannot be obtained (e.g., plugging in an unknown array)?  Is it possible to introduce "self-calibration" or "coordinate estimation" mechanisms in the future?

#### 5. **Impact of input audio duration and dynamic trajectory tracking**: How does the length of the input audio affect the model's overall performance? Is there a minimum continuous duration required for a sound source to yield reliable localization and reasoning results? Given that PhaseCoder operates on a 250ms analysis window and uses a 160ms hop size to align with the LLM, does this architectural choice impose a hard lower bound on the required audio length? Furthermore, within this specific time scale, is it currently possible to achieve dynamic tracking of a moving sound source's trajectory? While the paper acknowledges the assumption of static sources within the analysis window as a limitation, are there any preliminary results or observations on how the current frame-level embeddings fluctuate or perform when presented with continuously moving sources?

**Limitations:**

While the paper provides a solid evaluation across microphone counts, model sizes, and task types, the study would be significantly strengthened by a more systematic analysis of diverse acoustic environments. Specifically, exploring how sound field factors—such as reverberation time (RT60), near/far-field settings, and wall reflection coefficients—impact localization accuracy and spatial embeddings would provide deeper insights into the model's robustness. Expanding the current "free-floating" simulation to consider a physical device's acoustic baffling and diffraction effects would better reflect the complex challenges of real-world deployment.

**Strengths And Weaknesses:**

#### Strengths:
###### **Breakthrough in hardware limitations, achieving true microphone geometry independence**: It breaks free from the reliance of traditional spatial audio models (such as Ambisonics or specific beamformers) on fixed hardware, greatly enhancing the model's generalization capabilities across various edge devices.

###### **Simple and efficient architecture design**: It utilizes a pure Transformer backbone. By integrating sequential, frame, and microphone phase modulation positional embeddings, it ensures excellent performance while keeping the model lightweight (only about 6 million parameters).

###### **Enables general spatial audio reasoning capabilities for LLMs**: It innovatively introduces the concept of "spatial audio tokens" to accomplish four core tasks based on arbitrary microphone arrays: source localization, spatial reasoning, spatial transcription, and targeted transcription, greatly expanding the application prospects of multimodal LLMs in embodied intelligence.

---

> ### Author Rebuttal · Authors · 2026-03-30
>
> Thank you for insightful questions and comments as well as recognizing the contributions of this work.
>
> > Performance limits with very few microphones.
>
> The current setup will not work effectively with two microphones. With two omnidirectional microphones, there will be inherent front-back confusion. This confusion could be avoided by knowing head transfer related function (HRTF) or device-related transfer function. For example, we only have two ears, but localize any direction because our ears, head and shoulders have a unique directionality response.
> In the future work, we plan to see how to integrate HRTFs into the model.
>
> > Practicality of microphone coordinates and coordinate-less scenarios
>
> In the paper we assume that the microphone geometry is known in advance. This assumption works for many devices, as for example Android API provides metadata on locations of the microphones. Also, we believe often the microphone geometry information can be obtained by the LLM from the internet (e.g., manufacturer website or images). This could be packaged as an LLM tool call.
>
> However, we agree, in some cases the positions might not be known. We hypothesize it might be possible to figure out positions automatically, for example by sending a specific pulse and looking at what each microphone receives. However, we have not yet explored this topic in enough detail and this is a future work.
>
> > How does the length of the input audio affect the model's overall performance?
>
> The length of the input audio doesn’t affect model performance. It just has to be at least 250 ms, and can be any duration. The PhaseCoder model operates independently on 250 ms of audio with 160 ms hops. In contrast to RNN approaches, there is no memory between the hops. Our design choice was to keep the encoder light and simple, and assume that things such as source tracking and smoothing will be done on the LLM.
>
> > Is there a minimum continuous duration required for a sound source to yield reliable localization and reasoning results?
>
> As long as there is speech within the 250 ms context window. The 250 ms seems to provide a statistically significant amount of information for localization. We have not tested very short single word utterance, capturing those might require a shorter hop, which would cause a significant compute trade off.
>
> > Impact of input audio duration and dynamic trajectory tracking:
>
> To directly address this question, we conducted additional evaluations for dynamic source evaluation. We used Task 3 in LOCATA dataset that has a static microphone array and a moving human speaker. The results look promising and show that the model can localize moving sources with about 10 degree error. The azimuth angle over time plots also show consistent tracking of moving sources. We will include the data and trajectory plots in the revised paper.
>
> | Dataset | MAE Azimuth degrees (↓) |
> | :--- | :--- |
> | LOCATA Task 3 Eval | 9.73 |
> | LOCATA Task 3 Dev | 9.96 |

---

> > ### Author Rebuttal · Reviewer_NeYq · 2026-04-02
> >
> > Thank you for the detailed response. I suggest the authors integrate the analysis into the Discussion section to ensure the completeness of the presentation.

---

### Official Review · Reviewer_zSY6 · 2026-03-13

**Soundness:** 2
**Presentation:** 3
**Significance:** 3
**Originality:** 3
**Overall Recommendation:** 4
**Confidence:** 4

**Summary:**

This paper presents PhaseCoder, a spatial audio encoder designed to be agnostic to microphone geometry. The architecture processes STFT outputs combined with three types of positional embeddings and is trained on simulated data for azimuth/elevation prediction and distance estimation. The model is further aligned with a LLM (Gemma 3n) using a specialized spatial QA dataset. Evaluation results demonstrate improvements in localization and spatial audio understanding performance.

**Compliance With Llm Reviewing Policy:**

Affirmed.

**Final Justification:**

My main concerns were on experimental validation (lack of spatial baselines) and the computational part, which I misunderstood due to a clarity issue in the alignment between tokens. I think they are resolved through the rebuttal. I still think that the architecture of PhaseCoder itself lacks novelty, but since it is the first to propose incoporating microphone-array agnostic encoder to LLMs, it seems like a valuable contribution. So I changed the score to Weak Accept.

**Key Questions For Authors:**

1. Computational Efficiency: How does the computational cost of PhaseCoder compare to other spatial models like Spatial AST of BAT, when using the same number of channels (i.e., 2-channel setup)? Sequence length is a crucial factor for computation when integrating audio encoders with LLM.

2. Loss Function Rationale: What was the motivation for choosing Cross-Entropy Loss based on discretization over Mean Squared Error on continuous values for azimuth/elevation/distance prediction? Is there an empirical advantage to this choice?

3. Token Alignment: Could you provide a more detailed explanation of how the spatial tokens are aligned with the audio tokens when being integrated to LLM?

**Limitations:**

yes

**Strengths And Weaknesses:**

## Strengths
- Presentation: The paper is overall well-written and easy to follow.
- Originality: It introduces the first microphone-geometry-agnostic spatial audio LLM, addressing a key limitation in existing spatial audio LLM that are constrained to fixed microphone arrays.
- Performance: PhaseCoder achieves strong results on microphone-invariant localization benchmarks and supports multi-task formats, outperforming existing SOTA (GI-DOAEnet) that are limited to azimuth prediction.

## Weakness
- Limited Novelty in Architecture: The primary component for achieving microphone-agnosticism is the Microphone Positional Embedding. However, it is directly adapted from previous work (GI-DOA Net). This raises concerns regarding the technical novelty of the encoder design regarding its purpose.

- Extremely Long Sequence Length: The model generates 132 tokens/sec per channel, resulting in a significantly higher token rate than standard audio encoders for large audio-language models like HuBERT, BEATS, or Whisper (~50 tokens/sec). Compared to other spatial encoder like SpatialAST of BAT (which is based on AST and also use ~50 tokens/sec), the computational overhead of PhaseCoder seems excessive and may limit its scalability.

- Unclear Sequential Alignment: In Section 3.3, the authors state that PhaseCoder embeddings produce a sequence of spatial tokens at the same 160 ms hop size. The following example shows that spatial and audio tokens both consist of 188 tokens. However, given the multi-channel nature and much longer raw sequence length of PhaseCoder, it is unclear how this alignment is done. For instance, are the authors utilizing only a single CLS token for each 160ms window? More clarity on how these sequences are structured and interleaved is needed.

- Lack of Spatial Baseline Comparisons in LLM Evaluation: In Table 2, the proposed method is the only model capable of spatial audio processing. This makes it difficult to gauge the true significance of the performance. Even if a direct mic-agnostic competitor does not exist, the authors should have provided proxy baselines (e.g., a two-stage pipeline composed of a standalone localization model feeding output to an LLM) to contextualize the results.

---

> ### Author Rebuttal · Authors · 2026-03-30
>
> Thank you for the review and thoughtful feedback about our work!
>
> > Computational Efficiency
>
> We did not highlight it in the paper, but the PhaseCoder model was specifically designed to be **lightweight** and with the ability to **stream** in real-time. We have ported the model to TFlite to demo real-time inference. To make a fair compute comparison, three models were benchmarked for 10 sec sequence, since Spatial-AST was designed to operate on a 10 sec window globally.  We used  torch.profiler for inference speed and memory and fvcore for FLOPs and run models on one H100 GPU.
> The results in the table show that **PhaseCoder is more memory efficient and has lowest inference time** compared to other models. This is true for 2, 4, and 8 microphone configurations. While, PhaseCoder theoretically has a larger number of FLOPs than GI-DOAENet, it has significantly lower inference time, likely due to its pure transformer architecture. We will highlight this result in the paper.
>
> | # Mics | Models | # Params ↓ | RAM (MB) ↓ | FLOPS (G) ↓ | Inference time GPU / CPU (ms) ↓ | Architecture | Geometry agnostic |
> | :--- | :--- | :--- | :--- | :--- | :--- | :--- | :--- |
> | 2 Mics | **PhaseCoder (ours)** | 6 M | **405** | *8.50* | **2.24 / 6.181** | Transformer encoder | ✅ |
> | 2 Mics | Spatial-AST | 90 M | 808 | 70.30 | *2.96* / *10.08* | AST | 🚫 |
> | 2 Mics | GI-DOAEnet | 2M | 702 | **2.65** | 12.54 / 17.84 | hybrid | ✅ |
> | - | - | - | - | - | - | - | - |
> | 4 Mics | **PhaseCoder (ours)** | 6 M | **805** | 16.87 | **3.68 / 6.58** | Transformer encoder | ✅ |
> | 4 Mics | GI-DOAEnet | 2 M | 1021 | **4.65** | 16.62 / 18.44 | hybrid | ✅ |
> | - | - | - | - | - | - | - | - |
> | 8 Mics | **PhaseCoder (ours)** | 6 M | **1606** | 33.60 | **7.26 / 9.47** | Transformer encoder | ✅ |
> | 8 Mics | GI-DOAEnet | 2 M | 1695 | **8.67** | 15.82 / 19.16 | hybrid | ✅ |
>
> > Lack of Spatial Baseline Comparisons in LLM Evaluation
>
> We added a **zero-shot** baseline where we used a two-stage approach: 1) Run the PhaseCoder on the multichannel audio and extract a text data arrays of azimuth, elevation and distances over time with 160 ms hops. 2) Use the mono audio, text and instructions as a prompt to Gemma model without fine-tuning.
>
> The localization performance was better than random chance, but significantly worse than fine-tuned Gemma. Interestingly, WER was the highest (worst) among other baselines. In Task 3, the model consistently missed the second speaker. The results below will be added to table 2.
>
>
> | Model | Eval Dataset | Task 1: MAE ($^\circ$) Az | Task 1: MAE ($^\circ$) Elev | Task 1: MAE (m) Dist | Task 2: Acc (%) | Task 3: WER (%) Mean / Med | Task 3: MAE ($^\circ$) Az | Task 3: MAE ($^\circ$) Elev | Task 3: MAE (m) Dist | Task 4: Acc (%) | Task 4: WER (%) Mean / Med |
> | :--- | :--- | :--- | :--- | :--- | :--- | :--- | :--- | :--- | :--- | :--- | :--- |
> | Gemma (2-stage) | Synthetic | 55.32 | 37.13 | 0.81 | 57.61 | 60.42 / 85.71 | 55.20 | 33.51 | 0.85 | 35.74 | 48.49 / 54.17 |
> | Gemma (2-stage) | RSL2019 | 54.26 | 57.76 | 1.12 | 0.88 | 36.50 /100.0 | 36.49 | 16.98 | 0.86 | 34.18 | 70.23 / 57.14 |
>
> > Limited novelty in architecture.
>
> While the positional embedding is adapted from GI-DOAEnet, applying a pure Transformer to raw multichannel audio to extract microphone-agnostic representations is novel. It provides a simple and efficient model that can be expanded on by others.
>
> > Sequence Length & Token Alignment
>
> PhaseCoder does not generate 132 tokens/sec per channel. Because all microphone-dependent information is distilled into a single representation, PhaseCoder produces only a single [CLS] token every 160 ms, equating to roughly 6.25 tokens/sec. This is independent of the number of input channels.
>
> Regarding alignment: The spatial tokens are not interleaved with the audio tokens, as doing so would disrupt Gemma's native audio understanding. Instead, spatial tokens are appended after the audio tokens, framed by specific start/end tokens. Temporal alignment is learned implicitly during fine-tuning. For example, Task 1 requires the model to list speaker localizations in chronological order, forcing the LLM to correlate the temporal arrangement of the spatial embeddings with the audio sequence.  The resulting model did not swap the speakers, thus showing successful alignment.
>
>
>
>
>
> > Loss Function Rationale
>
>
> Our reasons for choosing categorical (cross-entropy loss) vs regression for localization:
> * The majority of  state-of-the-art approaches, especially related to spatial audio LLM interfacing (e.g., Spatial-AST) use categorical loss. Furthermore, our training recipe and backbone architecture follow the ViT and AST (Audio Spectrogram Transformer) paradigms, which use cross-entropy objectives.
> * We can directly compare PhaseCoder  to microphone-agnostic  GI-DOAENet, which uses categorical loss.
> * Real-world sound localization is probabilistic, making a statistical distribution of locations more fitting than an exact regression target.

---

> > ### Author Rebuttal · Reviewer_zSY6 · 2026-04-03
> >
> > Dear Authors
> >
> > Thank you for the detailed response. Computation analysis, additional baseline results, and clarification on alignment resolves most of my concern. I will raise my score.
> >
> > Best regards,
> > Reviewer zSY6

---

### Decision · Program_Chairs · 2026-04-30

**Decision:**

Accept (regular)

**Comment:**

This paper tackles an important problem and offers a practical method for bringing spatial audio understanding to multimodal LLMs without tying the system to a fixed microphone array. Reviewers generally agreed that the problem is important and that the empirical results are strong overall. The main initial concerns were about novelty, baselines, computational cost, token alignment, and the paper’s positioning compared to Ambisonics/beamforming front-ends that operate with known geometry.

The rebuttal addressed most of these issues by adding compute analysis, a two-stage baseline, a clearer discussion of token alignment, and a more explicit discussion of key limitations. Most reviewers were satisfied after discussion, though one reviewer remained unconvinced by the paper’s central novelty claim. Overall, I think the paper makes a solid contribution. The final version should incorporate the main clarifications from the rebuttal and present the paper’s positioning more carefully, especially by clarifying how the method differs in practice from geometry-dependent approaches.